# Biocompatibility Research of Magnetosomes Synthesized by *Acidithiobacillus ferrooxidans*

**DOI:** 10.3390/ijms26094278

**Published:** 2025-04-30

**Authors:** Bai-Qiang Wu, Jun Wang, Yang Liu, Bao-Jun Yang, Hui-Ying Li, Chun-Xiao Zhao, Guan-Zhou Qiu

**Affiliations:** 1School of Minerals Processing & Bioengineering, Central South University, Changsha 410083, China; wubaiqiang2025@126.com (B.-Q.W.); yangbaojun0312@126.com (B.-J.Y.); 18327518576@163.com (H.-Y.L.); chunxiao202502@163.com (C.-X.Z.); qgz@126.com (G.-Z.Q.); 2Key Lab of Bio-Hydrometallurgy of Ministry of Education, Changsha 410083, China

**Keywords:** bacterial magnetosome, *Acidithiobacillus ferrooxidans*, biocompatibility

## Abstract

Magnetosomes are magnetic nanocrystals synthesized by bacteria that have important application value in biomedicine. Therefore, it is very important to evaluate their biocompatibility. It has been reported that the extremophilic acidophilic bacterium *Acidithiobacillus ferrooxidans,* which is aerobic, can synthesize intracellular Fe_3_O_4_ magnetosomes. In this paper, we performed a comprehensive and systematic evaluation of the biocompatibility of magnetosomes with an average particle size of 53.66 nm from Acidithiobacillus ferrooxidans, including pharmacokinetics, degradation pathways, acute systemic toxicity, cytotoxicity, genotoxicity, blood index and immunotoxicity. The phase composition of the magnetosomes was identified as Fe3O4 through XRD and HRTEM analyses. Biocompatibility evaluation results showed that magnetosomes metabolized rapidly in rats and degraded thoroughly in major organs, with almost no residue. When the injection concentration was low (40 mg/kg, 60 mg/kg), magnetosomes would not cause pathological changes in the major organs of mice, basically. At the same time, magnetosomes had low cytotoxicity, genotoxicity, immunotoxicity and hemolysis rate, which proved that the magnetosomes synthesized by Acidithiobacillus ferrooxidans are magnetic nanomaterials with good biocompatibility. This research provides an important theoretical basis for the large-scale application of bacterial magnetosomes as functional magnetic nanomaterials.

## 1. Introduction

Magnetosomes were first identified as nanoparticles composed of iron oxide (Fe_3_O_4_) or iron sulfide (Fe_3_S_4_) under the strict genetic control of magnetotactic bacteria (MTB). They are formed through bacterial intracellular mineralization. Five operons strictly regulate the bacterial intracellular mineralization process, which is called the “magnetosome island” [1,2,3]. Bacterial magnetosomes (BMs) have the characteristics of uniform size, easy functional modification and high biocompatibility [4,5,6,7], and are a functional magnetic nanomaterial with potential application value in drug carriers, tumor hyperthermia, gene diagnosis, etc. [8,9,10]. In the size range of 10 nm–100 nm, the magnetosomes facilitate circulation in the system and infiltration into the target tissue through capillaries. The intrinsic magnetic properties facilitate the targeted drug delivery process [9]. They facilitate the release of pharmaceuticals to specific organs through magnetic targeting to particular tissues or robust ligand–receptor interactions [10], and can be conjugated with antibodies and chemotherapeutic agents [11]. In addition, magnetosomes are used as contrast agents in magnetic resonance imaging (MRI), for molecular imaging to diagnose disease, or as a novel enzyme immunoassay to detect immunoglobulin G [12,13]. On the other hand, magnetic nanoparticles pose potential risks, as they are nanoparticles that are separated from bacterial cells. Due to the insolubility of Fe_3_O_4_, the toxicity of the magnetic nanoparticles may mainly be caused by physical factors of the nanoparticles, which have been proven to lead to thrombosis, blockage and deposition within the body. Therefore, it is essential to evaluate the biocompatibility of purified magnetosomes and their possible applications in vivo. Magnetosomes have emerged as a promising candidate for biomedical applications. Among various nanoparticles, magnetosomes have received increasing attention due to their magnetic and guided drug delivery properties.

Biocompatibility refers to the performance of living tissue in reacting to an inactive material, generally referring to the compatibility between the material and the host. Nanomaterials can be applied only if they have good biocompatibility. Previous studies on the compatibility of bacterial magnetosomes have been reported. For example, Sun et al. obtained 62.7 mg/kg LD50 by injecting magnetosomes into the sublingual vein of SD rats, and briefly tested the immunotoxicity of magnetosomes [14]. The results showed that the immunotoxicity of magnetosomes was low. Recently, Haripriyaa et al. [11] coupled magnetosomes with fluorescein isothiocyanate (Mag-FITC) and studied the biological distribution of Mag-FITC in vivo with BALB/c mice at different time intervals through a bioimaging system. It was found that Mag-FITC caused neither death nor physical distress in the mice, and both were eliminated after 36 h of injection, leaving only a weak presence. Furthermore, metabolic and elimination analysis showed that the magnetosomes were fully metabolized within 48 h. In addition, some other researchers also concluded that BMs cause no histological damage and no abnormal biochemical parameters [15,16]. T. Revathy et al. [17] mainly evaluated the toxicity of magnetosomes in human red and white blood cells, a mouse macrophage cell line (J774), onion root tips and fish (Oreochromis mossambicus), with the magnetosomes only inducing low toxicity. These results show that magnetosomes are safe at lower concentrations and do not pose any potential risk to the ecosystem. In conclusion, many studies have shown that magnetosomes have good biocompatibility as functional magnetic nanomaterials.

However, because most of the magnetotactic bacteria (MTB for short) are anaerobic bacteria [18,19], it is difficult to carry out large-scale pure culture [20], so it is difficult to achieve the large-scale production of magnetosomes. Acidithiobacillus ferrooxidans (A. ferrooxidans for short) is a kind of aerobic, extremely acidophilic bacterium [21]. Meanwhile, A. ferrooxidans is also a Gram-negative, autotrophic, rod-like bacterium. When oxygen is sufficient, it can obtain energy by oxidizing Fe(II) and reducing inorganic sulfur compounds for bacterial growth. A. ferrooxidans plays an important role in the geochemical cycle [22,23]. As early as 2009, Liu Xinxing et al. [24] reported for the first time that A. ferrooxidans can mineralize into magnetic bodies within the cell. Since then, it has been found that four “magnetosome island” genes, such as mpsA, magA, thy and mamB, are associated with the intracellular mineralization of A. ferrooxidans into magnetosomes. Therefore, the magnetosome synthesized by A. ferrooxidans in the cell is considered to be special. It has also been reported that A. ferrooxidans has the potential to synthesize magnetosomes. The researchers believe that the use of ferrous oxide bacteria as a model strain could replace magnetotaxis for the mass production of magnetosomes.

Although several advantages of A. ferrooxidans magnetosomes have been mentioned previously, the pharmacokinetics, cytotoxicity, genotoxicity, immunogenicity and damage to tissues and organs of magnetosomes have not been studied in detail. Therefore, these parameters should be considered before their further recommendation, as they can deposit in the body and induce embolism [11]. Many future applications where magnetosomes could be used to treat or diagnose diseases also raise concerns about the potential risk of side effects from magnetosomes. It has been reported that after entering the circulatory system, the original nanoparticles come into contact with plasma proteins and blood cells and may initiate pathophysiological processes [25,26]. In this paper, L929 cells, mice and rats were used as research objects to evaluate the pharmacokinetics and in vivo degradation, acute systemic toxicity, cytotoxicity, genotoxicity, immunotoxicity and blood compatibility of BMs synthesized by A. ferrooxidans. Commercial magnetic nanomaterials (MNPs for short) were used as controls. This study will promote the clinical understanding and application of magnetosomes, which also has important academic significance for the biosafety research of bacterial magnetosomes, and provides an important theoretical basis for their large-scale application to brain metastases as functional magnetic nanomaterials.

## 2. Results

### 2.1. Bacterial Growth Curve

*Acidithiobacillus ferrooxidans* were sampled at different periods and counted by the blood cell count plate method to obtain their growth conditions (Figure 1). The concentration of *A. ferrooxidans* was 1.6 × 10^7^ cells/mL after a short lag period after inoculation. The bacteria began to enter logarithmic growth at about 6 h; the growth rate was the fastest from 24 h to 36 h, and the concentration of bacteria increased rapidly. The logarithmic phase of rapid bacterial growth lasted until about 48 h, and the highest bacterial concentration was 14.7 × 10^7^ cells/mL. After that, the bacteria remained in a stable state for 60 h. From 60 h to the end of culture, the number of bacteria decreased slightly, with the lowest concentration of 12.8 × 10^7^ cells/mL. The reason for the short maintenance time of the stable period in the culture process is that more inorganic energy substances are continuously consumed in the medium to maintain the steady expansion and growth of the bacterial community, and harmful metabolic substances are continuously accumulated during the active growth of bacteria in the logarithmic period, resulting in an unfavorable overall culture environment for bacterial growth.

### 2.2. XRD Analysis

The XRD analysis revealed that the diffraction pattern of magnetosomes synthesized intracellularly by *A. ferrooxidans* matched the standard diffraction peaks of Fe_3_O_4_ [27]. The characteristic peaks at 2θ = 30.46°, 35.35°, 43.68°, 54.33°, 57.13° and 63.75° corresponded to the (220), (311), (400), (422), (511) and (440) crystal planes, respectively. The full width at half maximum (FWHM) values of the magnetosome diffraction peaks were measured as 0.078° (220), 0.138° (311), 0.295° (400), 0.058° (422), 0.033° (511) and 0.093° (440) (Figure 2). These results demonstrate the well-defined crystalline structure of the magnetosomes biosynthesized by *A. ferrooxidans*.

### 2.3. TEM, STEM and HRTEM

*A. ferrooxidans* cells were sliced and then observed under TEM. The results showed that *A. ferrooxidans* can synthesize magnetosomes in an amorphous state in the cell, and the magnetosomes were freely distributed in the cell, distributed around the membrane intima; each cell contained 10 ± 2 magnetosomes (Figure 3A–C). There have been a number of studies on the synthesis process of magnetosomes based on biological transmission electron microscopy. Researchers have used transmission electron microscopy to characterize intracellular magnetosomes of magnetotactic bacteria and found that their magnetosomes form one or more magnetic links in the cell, and their shapes are mainly divided into four types: cuboid, approximately cuboid, bullet-shaped and near-spherical [28,29,30]. In comparison, the magnetic nanoparticles synthesized by *A. ferrooxidans* were closer to the amorphous shape and did not form magnetic linkage, which is obviously different from magnetotaxis bacteria living in an anaerobic alkaline environment.

STEM-EDXS element mapping and HRTEM were performed on the extracted magnetosomes to locate and characterize the magnetic nanoparticle phase. *A. ferrooxidans* forms irregular crystal particles in the cell, and the element composition is mainly iron and oxygen (Figure 3D–F). Fourier transform analysis of the lattice fringes by digital micrography shows that the crystal plane spacing of the lattice fringes is 2.92 A, which corresponds to the (220) crystal plane of magnetite, proving that the composition of the mentioned magnetic nanoparticles is Fe_3_O_4_ (Figure 3G,H).

### 2.4. Zeta Potential and Particle Size Analysis

Zeta potential and nanoparticle size (NanoSIMS) analyses are mainly used to characterize the size and stability of magnetic nanoparticles extracted from *A. ferrooxidans*. The average Zeta potential value of the intracellular magnetic nanoparticles of *A. ferrooxidans* is −41.5 ± 2.16 mV (Figure 4A); according to the standard of Zeta potential value, magnetic nanomaterials in this electric potential value belong to a relatively stable state. This proves that the intracellular magnetic nanoparticles of *A. ferrooxidans* do not have a trend of aggregation or agglomeration, and the state is relatively stable. At the same time, the nanoparticle size analyzer analysis found that the average particle size of the intracellular magnetic nanoparticles of *A. ferrooxidans* was about 61.22 ± 5.34 nm (Figure 4B). The polydispersion index (PDI) of the magnetosomes is 0.0872. Previously, Nan et al. [31] used the same method to measure the particle size of the intracellular magnetic nanoparticles of the magnetotactic bacterium MSR-1, and found that the average particle size was 53.66 nm, and the average Zeta potential value was −30.9 mV. Compared with this result, the intracellular magnetic nanoparticles of *A. ferrooxidans* have a larger particle size and higher Zeta potential value, which proves that the intracellular magnetic nanoparticles of *A. ferrooxidans* are more stable; Wu Lingbo et al. [32] used transmission electron microscopy to measure the particle size of the intracellular magnetic nanoparticles of *A. ferrooxidans* at about 41.5 ± 14.0 nm, and the results of this paper are larger than this.

### 2.5. BMs Pharmacokinetics

The pharmacokinetic properties of magnetic nanomaterial BMs were investigated after the injection of magnetic nanomaterial BMs into rats’ tail veins. The results showed that, as shown in Figure 5, the Fe contents in the blood of rats in the BM group and the MNP group were 22.00 ± 2 μg/mL and 21.0 ± 0.1 μg/mL 1 h after injection, and the Fe content in the blood continued to decline as time went on. At 24 h, the Fe content in the blood of the BM group decreased to 7.00 ± 0.09 μg/mL, and that of the MNP group was 6.41 μg/mL, respectively. By hour 72, the Fe content in the blood of the BM group had decreased to 4.70 ± 0.2 μg/mL, and that of the MNP group was 5.12 ± 0.10 μg/mL, respectively. These results are only 2.3% and 2.5% of the Fe content of the injection dose, so it can be seen that the BM magnetic nanomaterials have a short circulation time in the model organism and can be discharged in a short time.

At the same time, the half-life of Fe calculated by PKSolver 2.0 is 80.97 h (BMs) and 97.80 h (MNPs), respectively, and the half-life of Fe in BMs is shorter than that of MNPs. The rest of the pharmacokinetic parameters are shown in Table 1. We found that for the entire process of metabolism, the average retention time (MRT, indicating the time required to clear 63.2% of BMs and MNPs from the body) and other parameters, BMs were better than MNPs. We evaluated the blood drug concentration curve of the time axis surrounding area (AUC0-72-h, evaluation of BMs and the degree of absorption of MNPs), which appeared after the high blood drug concentration treatment (Cmax, reflecting the absorption rate and degree of BMs and MNPs in the body). This further indicates that BMs have superior retention and absorption properties compared to MNPs in vivo.

### 2.6. BMs In Vivo Degradation

BMs were injected into the tail vein of rats every 3 days. After 10 days, the rats were dissected, and livers, spleens and kidneys were taken from each group. The degradation of magnetic nanomaterials of BMs in the liver, spleen and kidney was analyzed by in situ XPS detection. The effect of BM clearance in vivo was further analyzed, with MNPs as a reference. As shown in Figure 6, there is no Fe in the liver, spleen and kidney, and the peak shape of C1s and O1s and the binding energy of each fission peak do not change in the high-resolution spectrum, indicating that the distribution of BMs in the liver, spleen and kidney may degrade, and no iron bicarbonate or carbonate is formed. This result proves that BMs had been excreted in mice. It also indicated that BMs degraded rapidly in vivo within the concentration range of the test, and they had good safety indications for major organs such as the liver, spleen and kidney. No Fe was detected in the MNP group.

### 2.7. Hematoxylin–Eosin Staining of Major Organs

Pathological examination results of major organ tissues (liver, spleen, kidneys) showed the following: After the injection of different concentrations of BMs into the tail vein of mice, for liver tissue, when the injection dose of BMs was 100 mg/kg, cytoplasmic light staining and occasional focal infiltration of portal inflammatory cells appeared in the liver tissue, and no abnormal pathological changes were observed at the other two lower injection doses. In the MNP group, when the injection dose was 40 mg/kg and 100 mg/kg, the liver tissue showed loose cytoplasmic staining and focal infiltration of rare inflammatory cells in the hepatic lobules (Figure 7A–H). For spleen tissue, except for a small part of white pulp in the BM 60 mg/kg group, no obvious pathological changes were found. However, white pulp was observed in all three dose groups of MNPs, especially in the high-dose group (100 mg/kg), where neutrophils were slightly increased, indicating that MNPs may have a mild pathological effect on the spleen of mice (Figure 7I–P). For kidney tissue, in the BM group, when the injection dose was 100 mg/kg, renal tubular epithelial cells in the medullary junction showed watery degeneration, cell swelling, loose cytoplasm and light staining, and no abnormal pathological changes were observed at the other two lower injection doses. In the MNP group, when the injection concentrations were 60 mg/kg and 100 mg/kg, the renal tubular epithelial cells showed watery degeneration, cell swelling, and loose and pale cytoplasm at the medullary junction (Figure 7Q–X).

In the BM group, no obvious abnormalities were observed in the liver, spleen, kidneys, heart, and lungs of mice when the injection dose was 40mg/kg body weight, and the tissue structures were normal. However, minor lesions were observed in the liver and spleen of the MNP-treated group. Other high-dose results show that both the BM and MNP groups exhibited minor lesions to varying degrees, but the BM group’s results were still better than those of the MNP group.

### 2.8. Proliferation of L929 Cells Detected by CCK-8

By calculating the cell proliferation rate (RGR) of L929, it can be found that, as shown in Figure 8A–C, the cell proliferation rate of BMs 0.5 mg/mL and the treatment group were 99.69 ± 0.017% in 24 h. When the treatment concentration was increased to 1 mg/mL, 2 mg/mL and 4mg/mL, the cell proliferation rate decreased to 97.45 ± 0.017%, 95.77 ± 0.016%, 93.42 ± 0.002%, but all were above 90%, and the cytotoxicity grade was grade 1. The cell proliferation rate of the MNP 0.5 mg/mL treatment group was 95.14 ± 0.016%, and when the treatment concentration was increased to 1 mg/mL, 2 mg/mL and 4 mg/mL, the cell proliferation rate decreased to 80.25 ± 0.045%, 70.73 ± 0.008% and 63.99 ± 0.007%, respectively. Except for the 0.5 mg/mL treatment group, the cell proliferation rate was significantly lower than that of the negative control group. The cell grade of the 2 mg/mL and 4 mg/mL treatment groups has been upgraded to grade 2. The same is true for 48 h and 72 h. It can be concluded that the cell proliferation rate of the BM-treated group under three different treatment times is significantly higher than that of the MNP-treated group, and the cytotoxicity of BMs formed by *Acidithiobacillus ferrooxidans* intracellular mineralization is lower than that of MNPs.

### 2.9. L929 Cells Total LDH Detection

In order to continue to evaluate the cytotoxicity of BM magnetic nanomaterials, L929 cells were treated with different concentrations of BMs for 24 h, 48 h and 72 h, and the cell membrane damage was evaluated by detecting the LDH release level, and then the cytotoxicity of BM magnetic nanomaterials was evaluated. As shown in Figure 8D–F, the LDH release values of the BM and MNP treatment groups showed significant differences compared with the negative control group after 24, 48 and 72 h of cell treatment (*p* < 0.01). However, at 48 h, when the treatment concentration of the BM group was 0.5 mg/mL, the LDH release value of the BM group was only 8.95 ± 0.077%, 10.35 ± 0.083% and 13.71 ± 0.043%, respectively. The release values were not high, and combined with the cell state diagram, the integrity of the cell membrane was not affected. In the MNP treatment group for 48 h, when the concentrations were 1 mg/mL, 2 mg/mL and 4 mg/mL, the LDH release values reached 19.60 ± 0.056%, 25.13 ± 0.068% and 30.39 ± 0.073%, respectively. Combined with the analysis of the cell state diagram, different degrees of membrane leakage may have occurred.

### 2.10. DNA Damage Detection

In this study, the genotoxicity of BMs to L929 cells was investigated by γ-H2AX immunofluorescence assay combined with classical mammalian erythrocyte micronucleus assay, and MNPs were used as a reference. The results showed that when the concentration of BMs or MNPs was high (i.e., 4 mg/mL), the content of γ-H2AX protein in each group was very low, and the color of green fluorescence pictures was light after 24 h and 48 h of cell culture, indicating that the nuclear DNA was not damaged (Figure 9 and Figure 10).

### 2.11. Blood Index 

The results of the blood routine test are often used to initially evaluate the toxicity of drugs. Therefore, we collected the blood of mice on the 4th day after injection of BMs and MNPs and evaluated whether BMs would cause inflammatory reactions in mice through blood routine tests (Figure 11). We focused on three indicators: white blood cells, red blood cells and platelets. Compared with the control group, the number of white blood cells (WBCs) and lymphocytes (LYMs) in the BM group was significantly different (*p* < 0.05), while the number of granulocytes (GRAs) was not significantly different (*p* > 0.05) (Figure 11A–C). The three indicators in the MNP group were similar to those in the BM group. The red blood cell-related indicators mainly include red blood cells (RBCs), hematocrit (HCT) and hemoglobin (HGB). After BMs entered the mouse body, there was no abnormal fluctuation in red blood cells and hemoglobin, and the hematocrit was slightly increased, but there was no significant difference compared with the control group. The indicators in the MNP group were lower than those in the BM group (Figure 11D–F). In addition, compared with the control group, the platelet (PLT) count, mean platelet volume (MPV), and platelet distribution width (PDW) in the BM group were not significantly different (*p* > 0.05); the platelet (PLT) value in the MNP group was significantly different (*p* < 0.05) (Figure 11G–I). This shows that BMs do not cause hemolysis and have good blood compatibility and low toxicity to the body, which is better than MNPs.

### 2.12. Complement Activation Level

In this study, ELISA was used to analyze the changes in the C3a level after BMs were incubated with rabbit serum for 60 min. The MNP group was used as a reference, and PBS and yeast A (1 mg/mL) were used as the negative control and positive control, respectively. Saccharin A is an insoluble carbohydrate from the yeast cell wall, which can activate rabbit serum complement and thus activate the complement cascade to produce an immune response, and it is especially used as a positive control in the immunoassay of immune proteins. When the concentrations of the BM group were 0.5 mg/mL and 1 mg/mL, the C3a level of the BM group was not significantly different from that of the negative control group (*p* > 0.05). When the concentration was 2 mg/mL, the level of C3a was significantly different from that of the negative control group (*p* < 0.05). When the concentration reached 4 mg/mL, the level of C3a was significantly lower than that of the negative control group (*p* < 0.01). For MNPs, there were significant differences in the C3a complement level between the MNP group and the negative control group in the experimental concentration range (*p* < 0.01) (Figure 12A).

### 2.13. Serum Cytokines Detection

We investigated whether an immune response is induced after the injection of BMs (4 mg mL^−1^) into the tail vein of mice. Serum levels of four important cytokines involved in the inflammatory response, including IL-2, IL-6, IFN-γ and TNF-α, were collected and detected on the 4th day of injection. The results are shown in Figure 12B–E: compared with the control group, BMs and MNPs did not cause significant changes in the expression levels of IL-2, IFN-γ and TNF-α. The expression level of IL-6 in the BM group was slightly down-regulated, but the difference was not significant compared with the control group (*p* > 0.05). The expression level of IL-6 in the MNP group was also down-regulated, and the difference was significant compared with the control group (*p* < 0.05).

## 3. Discussion

The main sources of *A. ferrooxidans* are ferrite and minerals, and *A. ferrooxidans* usually grows faster under iron (II) culture conditions, reaching the logarithmic phase within 48–72 h [22]. Under mineral (chalcopyrite, bornite, pyrite, etc.) culture conditions, *A. ferrooxidans* grows more slowly than under iron (II) culture conditions, but mineral conditions are more conducive to the acclimation of *A. ferrooxidans* in the bioleaching environment of sulfide ores, which improves its copper sulfide leaching efficiency [20,33]. In this paper, *A. ferrooxidans* reached the logarithmic phase within 12 h under iron (II) source culture conditions, growing faster than previously reported, which is presumed to be due to the fact that *A. ferrooxidans* had undergone multiple subcultures before this culture, and its cell proliferation rate had increased significantly compared to the initial cells. Therefore, this cultivation of *A. ferrooxidans* was successful, and the cultured *A. ferrooxidans* can be used for subsequent experiments.

The investigation on the phase of magnetosomes in *A. ferrooxidans* has been reported previously. Wu Lingbo et al. [32] used synchrotron X-ray absorption spectroscopy to prove that the mineral composition of magnetosomes synthesized by *A. ferrooxidans* ATCC23270 in vivo is Fe_3_O_4_. Yan Lei et al. [34] used XRD and HRTEM to prove that another strain of *A. ferrooxidans,* BY3, can synthesize Fe_3_O_4_ magnetosomes in vivo. These two results are consistent with the results of this study. However, the magnetosomes synthesized by *A. ferrooxidans* are amorphous. Therefore, based on the previous studies, it can be proved that the magnetosomes synthesized by *A. ferrooxidans* ATCC23270 in vivo are unique magnetosome crystals.

Pharmacokinetic characterization is an important stage to evaluate the biosafety and efficacy of nanomaterial carriers in drug delivery. M. Haripriyaa et al. [11] first combined fluorescein isothiocyanate (Mag-FITC) with the magnetic bodies of magnetic spirospira (MSR-1) and confirmed the combination by fluorescence microscopy. Then, BALB/c mice were used as subjects to detect the changes in iron concentration in the serum samples over different time periods. The results showed that the iron concentration gradually increased with the increase in time, and then gradually decreased with the further increase in time, indicating that the injected magnetic particles were being cleared from the bodies of mice. At the same time, iron can also be used by the body as a coenzyme, because it can directly carry out the REDOX cycle. Nan et al. [31] used Thermo ICAP–mass spectrometry to analyze the iron content in the major organs of mice after the intravenous injection of a magnetosome suspension. In the low-dose group (8 mg/kg, BMs), liver signals returned to normal levels by day 10, and the spleen quickly returned to normal by day 6. In the high-dose group (32 mg/kg, BMs), MRI signals in the liver and spleen of mice changed significantly from day one, indicating that BMs may accumulate mainly in these two organs, which may be because BMs infiltrate into the bile duct and further enter the duodenum with bile secretion. Over time, BMs are degraded into small particles and iron ions, resulting in a brief increase in iron levels in the spleen. Later, the degraded product is slowly excreted from the body through urination [35,36,37]. There were no significant MRI signal changes in the kidneys, indicating that BMs were not accumulating in the kidneys. These results are similar to those of our paper, which further indicates that BMs have the characteristics of easy degradation and will not cause serious toxicity to the body.

In a mouse systemic acute toxicity test, BMs had little effect on the growth of mice. The results of HE staining and the pathological examination of major organs of mice showed that no obvious abnormality was found in the liver, spleen, kidney, heart and lung at a higher dose of 40 mg/kg body weight, and the structure of each tissue was normal. According to a report by Nan et al. [31], when mice were injected with 8 mg/kg and 32 mg/kg magnetotactic bacteria magnetosomes, equivalent to 5 mg and 20 mg Fe/kg body weight, respectively, which are 10 times and 50 times the clinical dose, the liver and spleen tissues were collected and were stained with HE. No inflammatory cell infiltration, foam cell deposition, plaques, cellular degeneration or necrosis was found. Even at high doses of magnetosomes, very little residue was found in the heart, lung, kidney, and brain. Furthermore, all tissues showed normal appearance, indicating that magnetosomes caused minimal damage to organs. It was reported that M. Haripriyaa conducted HE staining and pathological section analysis on samples of the heart, liver, spleen, intestine, lung and kidney of rats treated with magnetic bodies, and the results showed that the microstructure of the tissues was normal, without signs of inflammation or fibrosis. No significant differences were observed in the tissue samples, and no histological abnormalities were detected [11]. It was again proved that magnetosomes have high biocompatibility, and the results also show that when magnetosomes are present in the system, they do not cause bioaccumulation or histological damage, and have been effectively removed from the system, leaving fewer trace elements. At the same time, we found that the main organ targeted by magnetosomes was the liver, followed by the spleen. Early studies reported that Fe_3_O_4_ nanoparticles were mainly taken up by reticuloendothelial cells abundant in the liver, such as some macrophages, and then distributed in the spleen, heart, lung and kidney [38]. The liver is the main organ where most drugs and exogenous substances are metabolized and removed. The hepatic lobules are the main structures of the liver that participate in the process of liver metabolism and the removal of nanoparticles [39].

So far, the cytotoxicity of bacterial magnetosomes has been studied in a variety of different mammalian cell lines, but the survival rates of different cell lines are significantly different. Nan Xiaohui et al. [31] evaluated the cytotoxicity of Raw 264.7 cells treated with magnetic bodies (BMs) and another commercial magnetic nanomaterial, PEG-Fe_3_O_4_ (MNPs), by the MTT method and LDH method, and the results showed that BMs had almost no cytotoxicity, and no significant difference was found from the negative control group. MNPs showed a significant difference (*p* < 0.01) compared with the negative control group, and showed certain cytotoxicity. Varalakshmi Raguraman [40] treated macrophages with BMs and MNPs, and the results showed a low cytotoxicity of BMs even at very high concentrations (250 µg/mL). MNPs showed a cell mortality rate of 8.9% at very low concentrations (10 µg/mL) and of 36.9% at 250 µg/mL. All the above studies showed that the cytotoxicity of BMs was lower than that of MNPs, which is basically consistent with the results of this paper.

For many envisioned applications of magnetosomes in vivo, their retention time in the body is usually quite low, and the nanoparticles are removed from the blood or degraded after a few hours. However, the biocompatibility provided by 24 h incubation may be useful not only for diagnostics, but also for potential in vivo applications. This study contributes to a comprehensive and clear understanding of the potential cytotoxic effects of magnetosomes. In addition, the results of LDH detection in this paper are different from those reported in the past. In this study, the LDH release value of the BM group was significantly different from that of the control group; the treatment concentration setting of the BM group was kept at the same order of magnitude as that of the CCK-8 experiment, reaching the milligram level (0.5–4 mg/mL), whereas the BM treatment concentration detected by LDH was all at the microgram level (10–250 µg /mL) in previous reports. In other words, the concentration of the BM treatment in this study was higher, so the research results are different.

DNA double-strand breaks are considered the most serious form of DNA damage. H2AX (H2A histone family member X) is a variant of the histone protein H2A. When the cell DNA double-strand breaks, ATM and ATR in the phosphatidylinositol 3 kinase-related kinase family can phosphorylate serine 139 on H2AX to form phosphorylated H2AX, that is, γ-H2AX. The level of γ-H2AX can clearly reflect the extent of DNA damage, so it is used in genotoxicity detection. In recent years, in vitro detection of γ-H2AX has gradually become a new method for evaluating in vitro genotoxicity, and an important marker for evaluating DNA damage response is phosphorylation of H2AX. In this study, γ-H2AX protein content was very low in all groups, indicating that nuclear DNA was not damaged. This result also proves once again that *A. ferrooxidans* does not cause nuclear DNA damage in the appropriate concentration range. In conclusion, BMs exhibit excellent cytocompatibility and reversible genotoxicity.

The blood compatibility of biomaterials is one of the important criteria to determine whether the biomaterials can be successfully applied in a clinical setting. The number of white blood cells directly reflects the inflammation of the body and is an important guardian of the body against inflammation. Red blood cells have many biological functions in the animal body, such as transporting oxygen and carbon dioxide, enhancing phagocytosis and the T cell-dependent response. Platelets are small pieces of cytoplasm cleaved from mature megakaryocytes in the bone marrow. Platelets are very important to the hemostatic function of the body. This study proves that BMs do not cause inflammation in the body, nor do they affect the process of red blood cells breaking down glucose to release energy. As early as 2010, it was reported that the blood routine test of mice treated with magnetotactic bacteria BMs found that, except for a slight abnormal increase in white blood cell indexes, other indexes were within the normal range, which is also consistent with the results of our paper [40]. In another study to assess the toxicity of magnetosomes in vivo, a blood test known as a complete blood count (CBC) was performed after intravenous administration of magnetosomes in mice. The CBC includes white blood cells, red blood cells and platelets, and it was performed after intravenous administration of magnetosomes in mice. Twenty-four CBC indexes of mice injected with BMs at different concentrations, from 0.5 mg/kg to 20 mg/kg, showed no significant differences from the control group, all of which were within the normal range, indicating once again that the magnetosomes have high blood compatibility [31].

If magnetosomes come into contact with blood during application, it is necessary to study whether they activate the complement system, which is one of the important links to evaluate their immunotoxicity. The C3 complement is the core of the complement system, and the complement pathway activated by it is the classical pathway. When the complement is activated, C3 is cut into C3a and C3b, and C3b either attaches to the nanoparticle or binds to a receptor on the surface of the immune cell, promoting the phagocytosis of the nanoparticle. The results of this study showed that the level of C3a complement in each experimental group was higher than that in the negative control group. On the one hand, as an antigen, magnetosomes stimulate the body to produce more antibodies in a very short time, and the antigen and antibody combine to produce more immune complexes, and after activating the complement, the serum complement level increases. On the other hand, since magnetosomes are injected directly into the blood circulation by intravenous injection and then enter tissues and organs more easily, more immune complexes are formed, and the amount of activated complement is also increased, so the level of active complement in the serum increases. However, MNPs activate the complement system more than BMs, and BMs are less immunotoxic than MNPs. In previous studies on the immunotoxicity of magnetosomes, Nan Xiaohui et al. [31] evaluated the complement activation levels of bacterial BMs and nanoscale PEG-MNPs, and found that BMs were more able to activate the complement system than MNPs. This study believed that the activation of magnetosomes might be due to BM membrane proteins and lipopolysaccharides, which are embedded in the purification process. These two substances are triggers for the activation of the complement system.

Cytokine interactions between receptor and target cells induce an immune response. Interleukin 2 (IL-2) is produced by activated Th1 cells, mainly stimulates the proliferation and differentiation of T and B cells and activates NK cells. Interleukin-6 (IL-6) has a wide range of cell sources, including T cells, B cells and macrophages, and its main function is to regulate the activity of B cells and T cells and participate in pro-inflammatory responses. Tumor necrosis factor (TNF-α), which is primarily produced by monocytes, macrophages and other cell types, serves as an effective mediator of inflammation and immune regulation, regulating the growth and differentiation of various cell types. Interferon (IFN-γ) is derived from Th1 cells and some CD8+T cells, and its main functions include promoting Th1 cell differentiation, stimulating IgG production and activating macrophages. The results of this study show that *A. ferrooxidans* intracellular BMs did not cause significant changes in the expression levels of IL-2, IFN-γ, IL-6 and TNF-α cytokines. Haripriyaa M et al. [11] tested the immunogenicity of magnetosomes in mouse serum samples, such as IL-2, IL-6, IL-8, IFN-γ and TNF-α, and found that the immunogenicity of all the pro-inflammatory markers expressed was less than 1%. These results suggest that these nanoscale structures do not cause embolism in later stages. The results in this paper are basically consistent with the reports, indicating that *A. ferrooxidans’* intracellular synthesis of magnetosomes within the appropriate concentration range will not produce an immune response.

## 4. Materials and Methods

### 4.1. Preparation of Bacterial Magnetosomes and Magnetic Nanoparticles

All chemicals were from Sinopharm Chemical Reagent Co. (Beijing, China) unless specified otherwise. The selected BM-producing strain was *Acidithiobacillus ferrooxidans* (ATCC 23270), which was stored and provided by the Biometallurgical strain storage Bank of the Key Laboratory of Biometallurgy, Ministry of Education, Central South University. 9K medium was used for culture, and bacterial growth was monitored. After being cultured, the *A. ferrooxidans* cells that had grown to the logarithmic phase were collected. Approximately 1 g of wet cell mass was suspended in 20 mL of Tris-HCl buffer (pH 8.8, concentration 1 M) and incubated in a water bath at 37 °C for 1.5 h, followed by intermittent ultrasonic treatment for 60 min under ice bath conditions (ultrasonic power: 50%, operation for 3 s, interval for 5 s). The treated cell suspension was then centrifuged at 2000 rpm for 5 min to collect the magnetosomes from the pellet. The supernatant was transferred into a flat-bottomed test tube and stored at 4 °C; partial precipitation of magnetosomes was observed after 2 days. Meanwhile, the centrifuged pellet was resuspended in 10 mM HEPES (pH 7.4) and allowed to stand on a magnet (S pole) for 30 min, after which the supernatant was discarded. This step (4) was repeated 8–10 times, and the extracted magnetosomes were preserved in physiological saline at 4 °C for subsequent use. Commercial Fe_3_O_4_ magnetic nanoparticles (MNPs) were used with a 50 nm Fe_3_O_4_ magnetic nanoparticle suspension, which was bought from the shiyanjia company, Changsha, China.

### 4.2. XRD Analysis

The XRD test of magnetosomes was carried out using an X-ray diffractometer (Bruker D8 advance, Changsha, China), with Cu-Kα radiation. The scanning range was from 20° to 80° (2θ angle), with 2°/min. The crystal phase composition of the magnetosomes was identified by analyzing the full width at half maximum (FWHM) of the corresponding diffraction peaks.

### 4.3. TEM Characterization

*Acidithiobacillus ferrooxidans* cells were sliced by a freezing microtome, and the result was then observed and imaged by a transmission electron microscope (Talos F200X) with an acceleration voltage of 80 kV from Changsha, China. The crystal structure and chemical composition of the magnetosomes were analyzed by HRTEM and energy dispersive X-ray spectroscopy (EDX), and by Talos F200X transmission electron microscopy with an accelerated voltage of 200 kV.

### 4.4. BM Extraction and STEM, HRTEM Characterization

*A. ferrooxidans* bacteria grown to the logarithmic stage were collected, and about 1g of wet bacteria were suspended in 20 mL Tris-HCI buffer (pH = 8.8, concentration 1 M) in a water bath at 37 °C for 1.5 h. Then, the treatment was performed in an ice bath for 60 min (ultrasonic power: 50%, running for 3 s, intermittent for 5 s). The treated cell suspension was centrifuged (2000 rpm, 5 min), the magnetic bodies in the precipitation were collected and then resuspended with 10 mM HEPES (pH7.4), and placed on the magnet (S pole) for 30 min. The supernatant was discarded, and this step was repeated 8–10 times. Finally, normal saline was added and preserved at 4 °C.

After extraction, they were stored in normal saline for high-resolution transmission electron microscopy. The high-resolution TEM study was carried out using a Talos F2000 at the Institute for the Advanced School of Central South University. The BMs were dispersed in ethanol by ultrasound and placed by analytical droplet on a double-layer membrane for study. High-resolution transmission electron microscopy (HRTEM), scanning transmission electron microscopy (STEM) and holographic diffraction analysis were performed at a 200 kV operating voltage in HAADF mode. Then, Digital Micrograph 3.5 was used to analyze the HRTEM images by fast Fourier transform (FFT) to study the crystal structure parameters.

### 4.5. Animal Experiments

Male Kunming rats (20 g ± 2 g) SPF, female SD rats (200 g ± 5 g) SPF and a New Zealand white rabbit (20.5 kg) were sourced from Changsheng Biotechnology Co., Ltd. (Shenyang, China). All animal experiments were approved by the Institutional Ethical Committee of Animal Experimentation of the Central South University.

### 4.6. Pharmacokinetics

Six female rats (about 200 g body weight, SPF grade) were randomly divided into two groups (*n* = 3), which were injected with BMs or MNPs (Fe quantitative, 6 mg/mL, 0.2 mL injection) through the tail vein, and blood was taken from the eyeball every 1, 8, 24, 48 and 72 h, and centrifuged at 3000 rpm for 10 min. Then, the serum was collected. The Fe content was detected by ICP-MS (Shiyanjia company, Changsha, China) after digestion at 180 °C, and the relevant pharmacokinetic parameters were calculated by PKSolver 2.0.

### 4.7. In Vivo Degradation Experiment

Six female SD rats were randomly divided into two groups (*n* = 3) and injected with BMs or MNPs (6 mg/mL, 0.2 mL) in the tail vein every 3 days, respectively. Ten days later, the rats in each group were dissected, and their livers, spleens and kidneys were taken, washed with normal saline, and then vacuum dried at 60 °C for 10 h, and ground into powder after complete drying. XPS detected the chemical morphology of BM or MNP degradation products in the liver, spleen and kidneys.

### 4.8. Acute Toxicity

The mice injected with BMs were randomly divided into three groups: BM group 1 (40 mg/kg weight), BM group 2 (60 mg/kg weight) and BM group 3 (100 mg/kg weight). A control group was injected with normal saline. After 1 week, the lightest-body-weight mice in each group were dissected, and the liver, spleen and kidney tissues were taken, respectively. The removed organs were washed with normal saline or PBS, then put into sample bottles and immersed in paraformaldehyde for preservation. Then, HE staining was performed, and pathological sections were analyzed. Hematoxylin–eosin staining was used for the HE staining.

### 4.9. Proliferation Detected by CCK-8

Cell culture plates with 96 wells were used to set up 9 treatment groups: a negative control group (DMEM medium), 4 BM treatment groups and 4 MNP treatment groups. After 24 h of L-929 cell plating, the original culture medium was discarded, and the culture medium containing BMs or MNPs was added (the final concentrations were 0.5 mg/mL, 1 mg/mL, 2 mg/mL and 4 mg/mL). After culturing for 24, 42 and 72 h, 10 μL of CCK-8 solution/well was added to the 96-well plate for 2 h, and the absorbance at 450 nm was measured by an enzyme labeler. Then, the relative cell proliferation rate (RGR (%) = the mean of the sample group/the mean of the negative control group × 100) was calculated.

### 4.10. Total Lactate Dehydrogenase (LDH) Activity Detection

After 24, 42 and 72 h of culture, the L929 cell culture plates were centrifuged with a perforated plate centrifuge. An amount of 400× *g* was centrifuged for 5 min, then 150 μL LDH was added to release the reagent, and the result was diluted 10 times with PBS; incubation continued for 1 h, and then 400× *g* was centrifuged for 5 min. An amount of 120 μL of supernatant was taken from each well and added to the 96-well plate. We added 60 μL LDH test solution to each well, mixed well, incubated the samples for 30 min, and then measured the absorbance at 490 nm. Cell mortality (%) = (treated sample absorbance − sample control hole absorbance)/(Maximum cell enzyme activity absorbance − sample control hole absorbance) × 100.

### 4.11. DNA Damage Detection

The nuclear DNA damage was analyzed by γ-H2AX immunofluorescence assay. The negative control group (DMEM medium), BM group and MNP group were set up, and the final concentration of the BM and MNP solution was 4 mg/mL. L-929 cells were cultured for 24 h and discarded after 48 h. The cells were fixed with 1 mL 4% paraformaldehyde for 15 min and cleaned with PBS. One mL 0.5% Triton X-100 was permeated at room temperature for 10 min, and the cells were cleaned with PBS. One mL3% BSA was enclosed with the samples for 30 min, and the cells were soaked with PBS. Then, 0.5 mL anti-γ-H2AX antibody (1:200) was incubated at room temperature for 2 h, and the cells were cleaned with PBS. Then, the cells were incubated at room temperature for 1 h with 1 mL FITC fluorescent-coupled secondary antibody (1:200, Bioss, Woburn, MA, USA) and cleaned with PBS. The nuclei were then stained with DAPI at room temperature for 5 min and soaked with PBS. Finally, cellular immunofluorescence images were observed under a fluorescence microscope (AX10 Carl Zeiss, Jena, Germany). Gamma-h2ax staining is green fluorescent, and nuclear DAPI staining is blue fluorescent.

### 4.12. Blood Index Detection

The experiment was divided into a control group (normal saline), BM group and MNP group. Three Kunming mice in each group were injected with BMs or MNPs in the tail vein; the dose was 0.1 mL (4 mg mL^−1^), and after 4 days, the blood of mice in each group was collected with a heparin anticoagulant tube. Normal saline was used as the control, and blood routine analysis was performed by an automatic blood analyzer.

### 4.13. Complement Activation Experiment

A BM or MNP solution was prepared with concentrations of 0.5 mg/mL, 1 mg/mL, 2 mg/mL and 4 mg/mL, respectively. PBS and yeast A (1 mg/mL) were used as the negative control and positive control, respectively. Rabbit blood was collected and placed in a 2 mL centrifuge tube. Serum was obtained by centrifugation 4000× *g* for 20 min after natural coagulation. Fresh rabbit serum in the amount of 0.5 mL was added to each group (0.5 mL), and incubation was performed at 37 °C for 60 min. The test tubes of each group were removed and placed in an ice bath to prevent further activation of the complement. The operation was performed according to the rabbit C3a enzyme-linked immunoassay (ELISA) kit, and the absorbance value was determined at the 450 nm wavelength. According to the absorbance value, the content of C3a in the BM group, MNP group, positive control group and negative control group was calculated.

### 4.14. Serum Cytokine Levels Assessment

Four serum cytokine levels were measured, including interleukin-2 (IL-2), interleukin-6 (IL-6), interferon gamma (INF-γ), and tumor necrosis factor (TNF-α). Cytokine detection was performed by a double-antibody one-step sandwich enzyme-linked immunosorbent assay (ELISA). The experiment was set up in four groups with 3 mice in each group. The blood of mice with BMs or MNPs (4 mg mL^−1^, 0.1 mL) injected into the tail vein for 4 days was collected, and the supernatant was centrifuged and stored at −80 °C. The cytokine content was detected according to the ELISA kit instructions. Mice were injected with normal saline as a control group.

### 4.15. Statistical Analysis

Experimental data are expressed as mean ± standard deviation (SD). We used origin 2021 software, and the differences between different groups were analyzed by paired or unpaired *t*-tests. When *p* < 0.05, it was statistically significant.

## 5. Conclusions

In summary, this study conducted a comprehensive analysis of the biocompatibility of Fe_3_O_4_ magnetosomes extracted from *A. ferrooxidans*, including pharmacokinetics, in vivo degradation, systemic acute toxicity, cytotoxicity, blood tests and immunotoxicity. Only in terms of systemic acute toxicity, high doses of magnetosomes (60 mg/kg and 100 mg/kg body weight) had minor pathological effects on major organs in mice, but this concentration far exceeded the clinical dose. Our results show that the magnetosomes synthesized by *A. ferrooxidans are* safe in many ways, and these results will provide a theoretical basis for the future improvement and better application of magnetosomes in clinical applications.

## Figures and Tables

**Figure 1 ijms-26-04278-f001:**
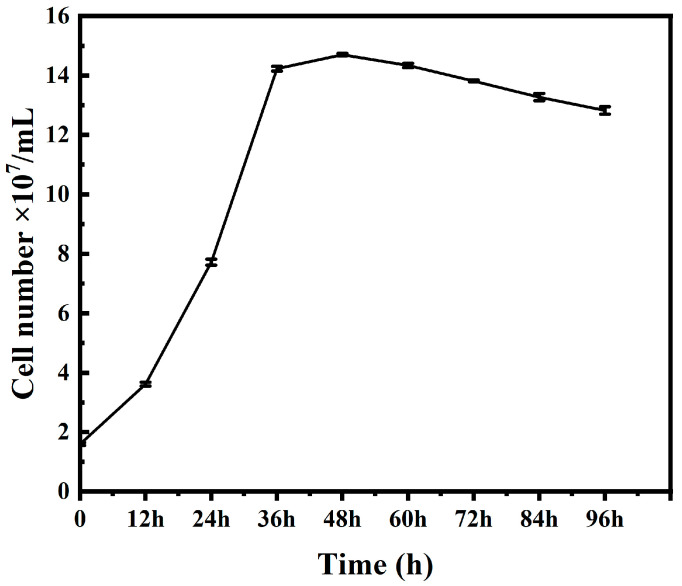
*A. ferrooxidans* growth curve.

**Figure 2 ijms-26-04278-f002:**
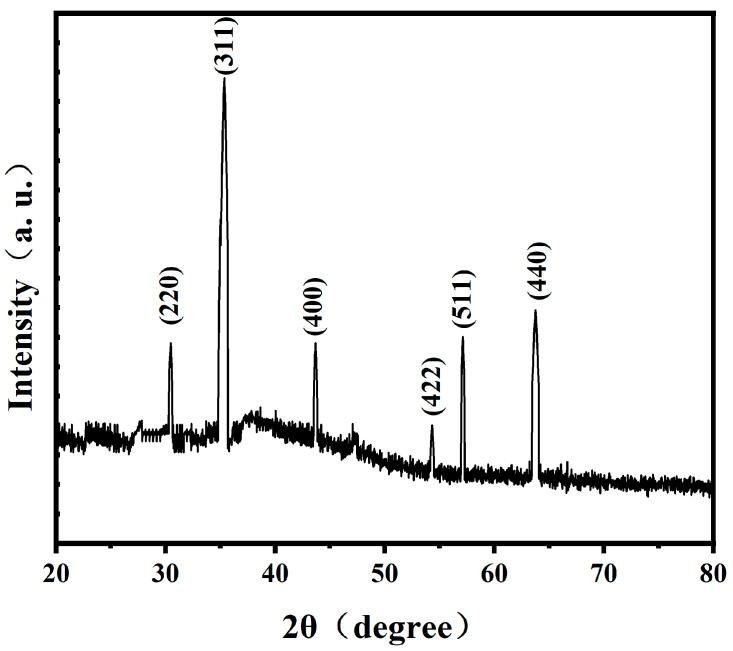
X-ray diffraction curves of magnetosomes synthesized by *A. ferrooxidans*.

**Figure 3 ijms-26-04278-f003:**
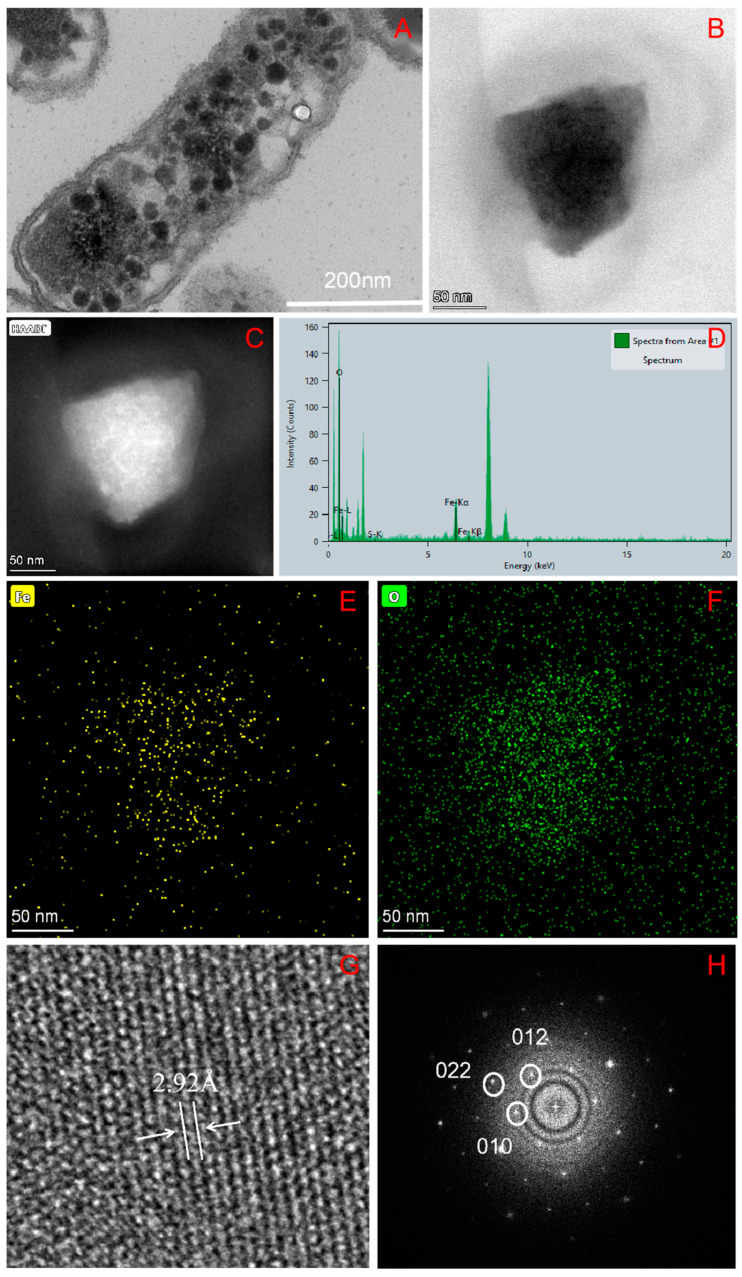
TEM, HETEM and EDS analysis of *A.ferrooxidans* intracellular magnetosomes (**A**): low-resolution transmission electron microscopy (TEM) image of *A. ferrooxidans* intracellular magnetosomes; (**B**): HRTEM image of magnetosomes; (**C**): Dark field HRTEM image of magnetosomes; (**D**): EDX spectrum of Fe and O; (**E**): Fe elemental mapping, (**F**): O elemental mapping, (**G**): Lattice fringe of magnetosomes, (**H**): Fourier Transform.

**Figure 4 ijms-26-04278-f004:**
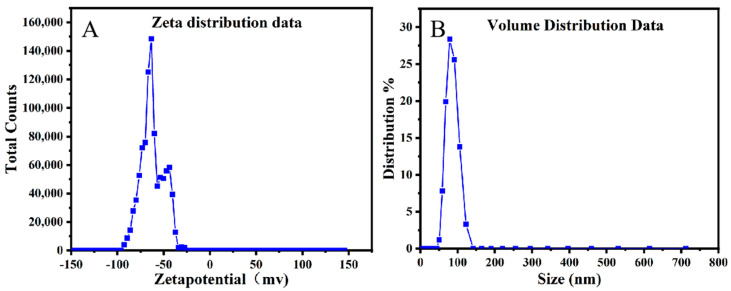
Zetapotential diagram and nanometer size analysis diagram: (**A**) Zeta potentiogram, (**B**) nanometer size analysis diagram.

**Figure 5 ijms-26-04278-f005:**
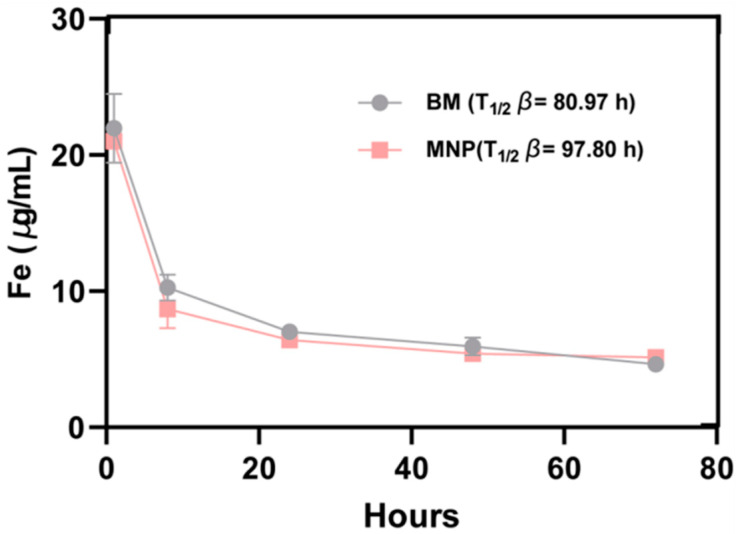
The pharmacokinetic curves of BMs and MNPs, and the concentration of Fe in the blood after the injection of BMs and MNPs in the tail vein for different time periods.

**Figure 6 ijms-26-04278-f006:**
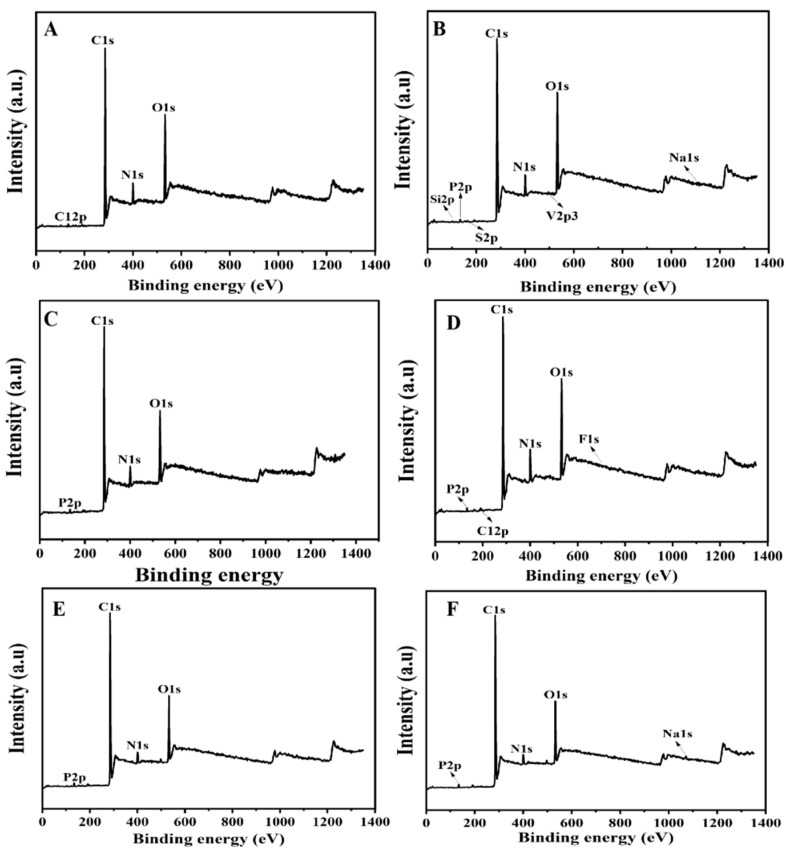
Full spectrum of XPS after degradation of BMs and MNPs in liver, spleen and kidney. (**A**): BMs in liver; (**B**): MNPs in liver; (**C**): BMs in spleen; (**D**): MNPs in spleen; (**E**): BMs in kidney; (**F**): MNPs in kidney.

**Figure 7 ijms-26-04278-f007:**
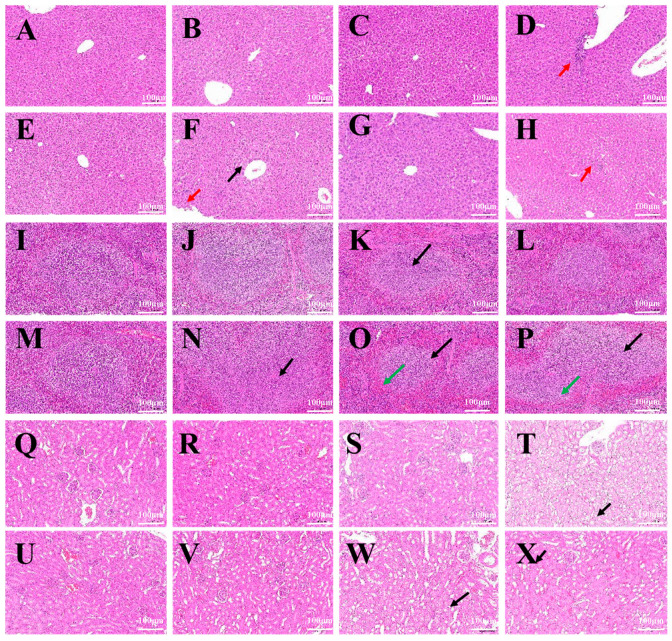
HE staining and pathological analysis of liver, spleen and kidneys. Liver (**A**–**H**). (**A**,**E**): control; (**B**): BMs 40 mg/kg; (**C**): BMs 60 mg/kg; (**D**): BMs 100 mg/kg (red arrow points to the focal infiltration of rare inflammatory cells in the hepatic lobules); (**F**): MNPs 40 mg/kg; (**G**): MNPs 60 mg/kg; (**H**): MNPs 100 mg/kg. Spleen (**I**–**O**). (**I**,**M**): control; (**J**): BMs 40 mg/kg; (**K**): BMs 60 mg/kg; (**L**): BMs 100 mg/kg; (**M**): MNPs 40 mg/kg; (**N**): MNPs 60 mg/kg; (**O**): MNPs 100 mg/kg. Kidneys (**P**–**W**). (**Q**,**U**): control; (**R**): BMs 40 mg/kg; (**S**): BMs 60 mg/kg; (**T**): BMs 100 mg/kg; (**V**): MNPs 40 mg/kg; (**W**): MNPs 60 mg/kg; (**X**): MNPs 100 mg/kg. (The red arrows point to the focal infiltration of rare inflammatory cells in the hepatic lobules. The black arrows point to the loose cytoplasmic staining. The green arrows point to the small part of white pulp).

**Figure 8 ijms-26-04278-f008:**
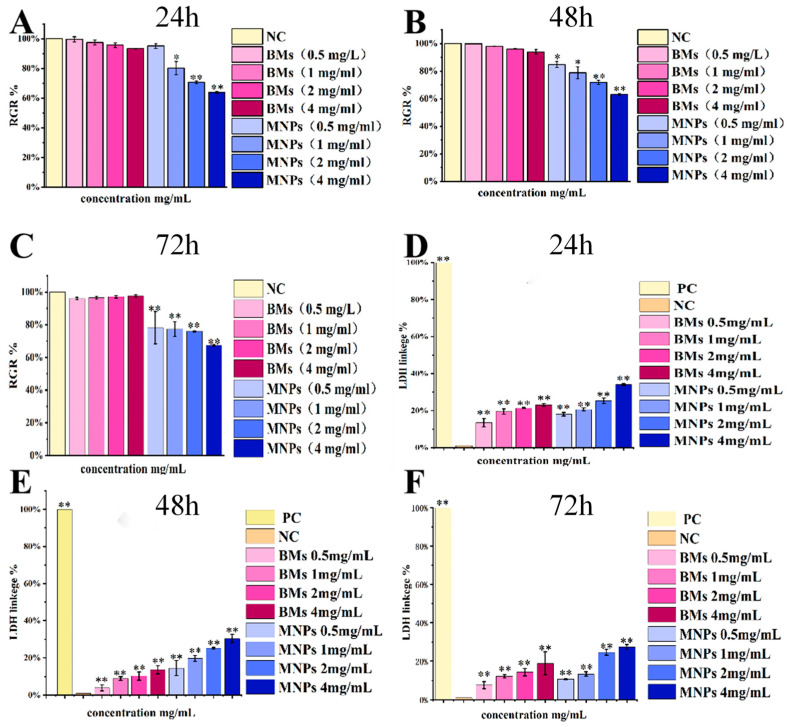
L929 cell proliferation rates and total LDH release values were detected at different times and concentrations of BM or MNP treatment. (**A**–**C**) RGR% of L929 cell—(**A**): 24 h BM and MNP L929 cell RGR%; (**B**): 48 h BM and MNP L929 cell RGR%; (**C**): 72 h BM and MNP L929 cell RGR%. (**D**–**F**) L929 cell LDH release value—(**D**): 24 h BM and MNP LDH release value; (**E**): 48 h BM and MNP LDH release value; (**F**): 72 h BM and MNP LDH release value. Untreated cells were used as the negative control (NC), and the absorbance of maximum enzyme activity of cells was the positive control (PC), * *p* < 0.05, ** *p* < 0.01 indicates significant difference from the negative group.

**Figure 9 ijms-26-04278-f009:**
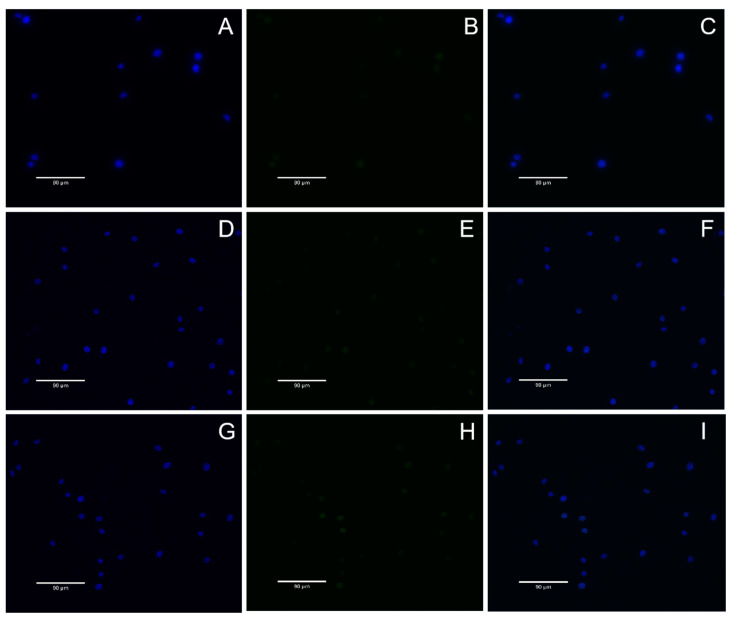
Negative control and BMs γ-H2AX immunofluorescence diagram. (**A**–**C**): Negative control group, (**A**): DAPI, (**B**): H2AX, (**C**): Merged; (**D**–**F**): BMs 24 h, (**D**): DAPI, (**E**): H2AX, (**F**): Merged; (**G**–**I**): BMs 48 h (**G**): DAPI, (**H**): H2AX, (**I**): Merged. DAPI refers to nuclear staining images, where the nucleus is stained with blue fluorescence; H2AX refers to the stained image of the target protein H2AX, which is stained with green fluorescence; Merged refers to the merge of the first two images.

**Figure 10 ijms-26-04278-f010:**
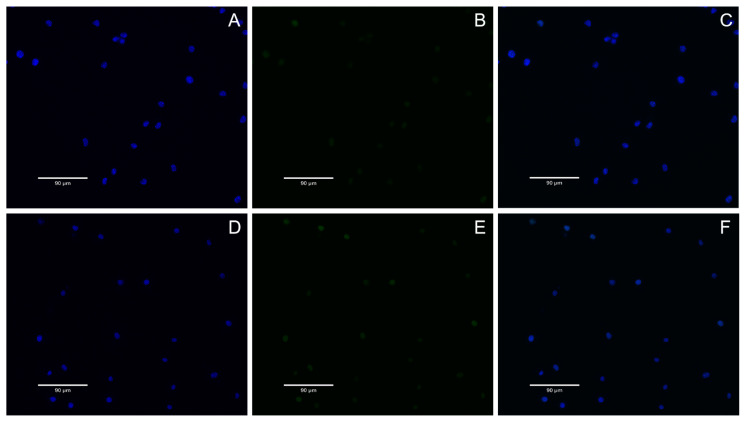
MNPs γ-H2AX immunofluorescence diagram. (**A**–**C**): MNPs 24 h, (**A**): DAPI, (**B**): H2AX, (**C**): Merged; (**D**–**F**): MNPs 48 h, (**D**): DAPI, (**E**): H2AX, (**F**): Merged. DAPI refers to nuclear staining images, where the nucleus is stained with blue fluorescence; H2AX refers to the stained image of the target protein H2AX, which is stained with green fluorescence; Merged refers to the merge of the first two images.

**Figure 11 ijms-26-04278-f011:**
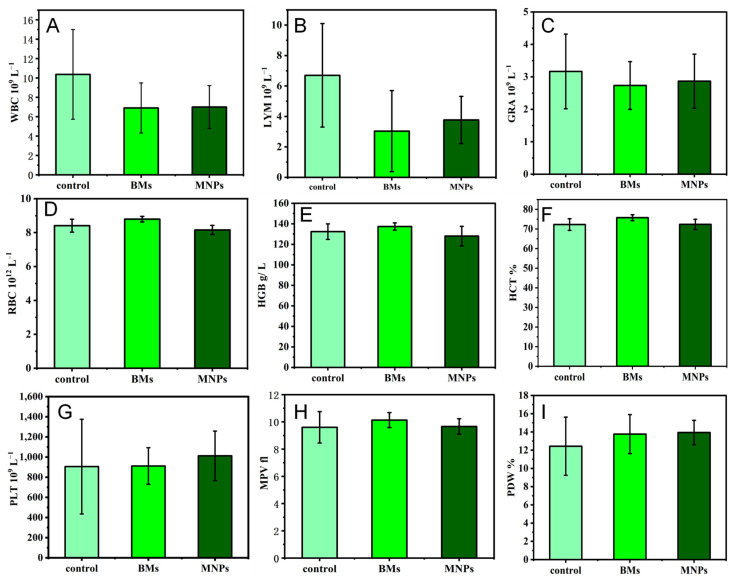
Blood routine test results on day 4 after caudal vein injection of BMs or MNPs. (**A**): White blood cells (WBCs); (**B**): lymphocytes (LYMs); (**C**): granulocytes (GRAs); (**D**): red blood cells (RBCs); (**E**): hemoglobin (HRC); (**F**): erythrocyte specific volume (HCT); (**G**): platelets (PLTs); (**H**): mean platelet volume (MPV); (**I**): platelet distribution width (PDW).

**Figure 12 ijms-26-04278-f012:**
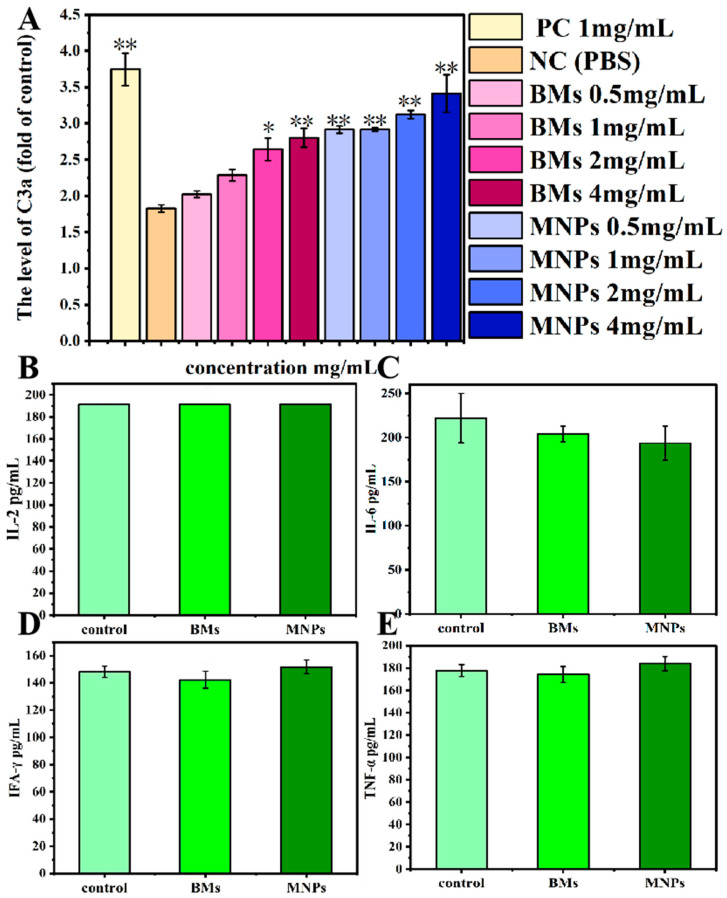
C3a complement activation level and serum cytokines detected with BM treatment. (**A**): ELISA detected C3a complement activation after incubation with BMs or MNPs (0.5 mg/mL, 1 mg/mL, 2 mg/mL, 4 mg/mL). PBS and yeast A (1 mg/mL) were used as negative control and positive control, respectively. * *p* < 0.05, ** *p* < 0.01 indicates significant difference from negative control. (**B**–**E**): Serum cytokines detected by ELISA on day 4 of caudal vein injection of BMs or MNPs. (**B**): IL-2 expression level; (**C**): IL-6 expression level; (**D**): expression level of IFA-γ; (**E**): expression level of TNF-α.

**Table 1 ijms-26-04278-t001:** Pharmacokinetic parameters of BMs and MNPs.

Parameter	BMs	MNPs
Half-life (h)	80.97	97.80
Maximum Concentration (μg/)mL	21.96	20.99
Mean Residence Time 0–∞(h)	108.08	137.03
Clearance ((mg)/(μg/)mL/h)	0.0011	0.000993

## Data Availability

The authors confirm that the data supporting the findings of this study are available within the article.

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
