# Peer review of "Biocompatibility Research of Magnetosomes Synthesized by Acidithiobacillus ferrooxidans"

_ijms, 2025, doi:10.3390/ijms26094278_

Round 1

Reviewer 1 Report

Comments and Suggestions for Authors

Line 34, "BMs have the characteristics..." It is imperative that the definition of "BMs" (bacterial magnetosomes?) be provided in order to ensure the clarity and comprehensibility of the text. To that end, all abbreviations should be expanded upon their initial appearance in the text of the manuscript.

Line 225 states that "the particle size is about 40 ± 6 nm (Figure 2.A, B)." It is necessary to clarify whether the figure in question depicts a solitary magnetosome or an agglomerate of magnetosomes within a bacterial cell, as the scale bar in the figure indicates a size of object greater than 100 nm. In the event that the former is depicted, inquiries emerge with respect to the dimensions. Additionally, Line 273 asserts that "the nanoparticle size analyzer analysis found that the average particle size of the intracellular magnetic nanoparticles of A. ferrooxidans was about 61.22 nm (Figure 3. B)." This value contradicts the claim made in Line 16 of the Abstract, which states that "the average particle size was 53.66 nm." Concurrently, the extant literature cites the isolation of nanoparticles measuring the same 53.66 nm by Nan et al. [30] through the utilization of the magnetotactic bacterium MSR-1 (line 287). Moreover, as the authors further expound, "In comparison with this result, the intracellular magnetic nanoparticles of A. ferrooxidans exhibit a larger particle size" (line 288). It is imperative to ascertain the discrepancy between these values and the true measurements.

It is imperative to reiterate the necessity of fully expanding all abbreviations. The intended meaning of the abbreviation "Cl_obs" in the Table 1 is not readily discernible. It is necessary to ascertain whether it signifies the observed clearance rate or if it is an abbreviation of another term.  

The Discussion section should contain a summary of the results obtained, and all additional references to previous related studies should be used solely for the purpose of discussion, if they confirm or refute the results obtained. The general descriptive part (e.g., the studied object or research methods, etc.) should be disclosed in the Introduction section. Therefore, it is recommended to review the Discussion section by moving non-essential information (e.g., lines 478-502, etc.) to the Introduction.

In summation of the aforementioned review, it is evident that the trajectory of research selected by the authors appears to be both intriguing and timely. It is foreseeable that this research will find practical application in both scientific and applied biomedical contexts in the imminent future. With minor adjustments, the presented work is deemed suitable for publication in this journal.

----------------

The following Suggestions may be of interest to the authors; however, they are not related to the presented work and should not be considered part of the aforementioned review. In future work involving bacterial-mediated synthesis, it would be worthwhile to attempt to dope magnetite nanoparticles with oxides of other metals that are promising for biomedical applications. For example, nanoparticles of rare earth metal oxides seem extremely promising at present. It is noteworthy that A. ferrooxidans bacteria have the capacity to leach La and Ce; in the presence of excess iron, they may potentially form magnetosomes of mixed composition. The hypothesis posits that the presence of cerium (or other light lanthanide) species in growth media will likely facilitate enhanced bacterial growth and activity and reduced magnetosome toxicity. Conversely, gadolinium species may potentially augment the MRI response of nanoparticles, etc. It is my sincerest wish that the authors of this work may encounter success in their future endeavors.

Comments on the Quality of English Language

The experimental material exhibits a high scientific level; however, the presentation's deficiency in English language proficiency diminishes its informative capacity. The presentation's inaccuracies become evident from the Introduction's opening sentence, which states, "Magnetosomes were first identified as nanoparticle properties that synthesize iron oxide (Fe3O4) or iron sulfide (Fe3S4)...". The object itself cannot be identified as a property; it is also unclear who (or what) synthesizes iron species under bacterial control. As illustrated in line 136 (Analytical droplet on a double-layer membrane for study.), there are instances of inconsistent sentence structure, characterized by the absence of a verb (predicate) in the sentence. To ensure the coherence and logical soundness of the text, a thorough review by the authors is imperative, accompanied by the rectification of all logical and syntactic errors. The optimal solution would be to have the text proofread by a native English speaker!

Author Response

Response to Reviewer 1 Comments

  1. Summary

Thank you for your comments. Through careful thinking, we try our best to answer your suggestions and comments. All revisions of comments and suggestions have been marked red in the revised manuscript. Meanwhile, all revisions of the quality of English language has been marked green in the revised manuscript.

  1. Questions of General Evaluation

Open Review

( ) I would not like to sign my review report

(x) I would like to sign my review report

Quality of English Language

(x) The English could be improved to more clearly express the research.

( ) The English is fine and does not require any improvement.

Yes Can be improved Must be improved Not applicable

Does the introduction provide sufficient background and include all relevant references?

(x) ( ) ( ) ( )

Is the research design appropriate?

(x) ( ) ( ) ( )

Are the methods adequately described?

(x) ( ) ( ) ( )

Are the results clearly presented?

( ) (x) ( ) ( )

Are the conclusions supported by the results?

(x) ( ) ( ) ( )

  1. Point by point response to Comments and Suggestions for Authors

Comment 1: Line 34, "BMs have the characteristics..." It is imperative that the definition of "BMs" (bacterial magnetosomes?) be provided in order to ensure the clarity and comprehensibility of the text. To that end, all abbreviations should be expanded upon their initial appearance in the text of the manuscript.

Response 1:Thanks for your constructive comments, the definition of "BMs" was imperatived in the Line 35 of the revised manuscript.

Comment 2: Line 225 states that "the particle size is about 40 ± 6 nm (Figure 2.A, B)." It is necessary to clarify whether the figure in question depicts a solitary magnetosome or an agglomerate of magnetosomes within a bacterial cell, as the scale bar in the figure indicates a size of object greater than 100 nm. In the event that the former is depicted, inquiries emerge with respect to the dimensions. Additionally, Line 273 asserts that "the nanoparticle size analyzer analysis found that the average particle size of the intracellular magnetic nanoparticles of A. ferrooxidans was about 61.22 nm (Figure 3. B)." This value contradicts the claim made in Line 16 of the Abstract, which states that "the average particle size was 53.66 nm." Concurrently, the extant literature cites the isolation of nanoparticles measuring the same 53.66 nm by Nan et al. [30] through the utilization of the magnetotactic bacterium MSR-1 (line 287). Moreover, as the authors further expound, "In comparison with this result, the intracellular magnetic nanoparticles of A. ferrooxidans exhibit a larger particle size" (line 288). It is imperative to ascertain the discrepancy between these values and the true measurements.

Response 2:Thanks for your constructive comments, we have modified the “the average particle size was 53.66 nm” to “the average particle size was 61.22 nm” and deleted the that "the particle size is about 40 ± 6 nm (Figure 2.A, B) in the revised manuscript. Furthermore, it should be noted that the study by Nan et al. cited in the manuscript focused on magnetotactic bacteria inhabiting anaerobic alkaline environments, whereas Acidithiobacillus ferrooxidans thrives in extreme acidic conditions. The magnetosomes synthesized intracellular by these two distinct bacterial species exhibit notable differences. Specifically, there is no direct correlation in terms of particle size between them. We have addressed this point in the revised manuscript by providing additional clarification in Results 3.3.

Comment 3: It is imperative to reiterate the necessity of fully expanding all abbreviations. The intended meaning of the abbreviation "Cl_obs" in the Table 1 is not readily discernible. It is necessary to ascertain whether it signifies the observed clearance rate or if it is an abbreviation of another term.  

The Discussion section should contain a summary of the results obtained, and all additional references to previous related studies should be used solely for the purpose of discussion, if they confirm or refute the results obtained. The general descriptive part (e.g., the studied object or research methods, etc.) should be disclosed in the Introduction section. Therefore, it is recommended to review the Discussion section by moving non-essential information (e.g., lines 478-502, etc.) to the Introduction

Response 3: Thank you for your valuable comments. We have expanded all abbreviations,especially for the pharmacokinetics parameters. Meanwhile, we have revised the Discussion section by deleting non-essential information, especially for the lines 478-502.

Comment 4 In summation of the aforementioned review, it is evident that the trajectory of research selected by the authors appears to be both intriguing and timely. It is foreseeable that this research will find practical application in both scientific and applied biomedical contexts in the imminent future. With minor adjustments, the presented work is deemed suitable for publication in this journal.

The following Suggestions may be of interest to the authors; however, they are not related to the presented work and should not be considered part of the aforementioned review. In future work involving bacterial-mediated synthesis, it would be worthwhile to attempt to dope magnetite nanoparticles with oxides of other metals that are promising for biomedical applications. For example, nanoparticles of rare earth metal oxides seem extremely promising at present. It is noteworthy that A. ferrooxidans bacteria have the capacity to leach La and Ce; in the presence of excess iron, they may potentially form magnetosomes of mixed composition. The hypothesis posits that the presence of cerium (or other light lanthanide) species in growth media will likely facilitate enhanced bacterial growth and activity and reduced magnetosome toxicity. Conversely, gadolinium species may potentially augment the MRI response of nanoparticles, etc. It is my sincerest wish that the authors of this work may encounter success in their future endeavors.

Response 4: We are profoundly grateful for your recognition and positive evaluation of this paper, which we consider a great honor. Furthermore, we are highly interested in the suggestions you have proposed for future research endeavors. In the subsequent phase, we intend to conduct a comprehensive review of the relevant literature and investigate the research on rare earth metal-doped magnetosomes. We eagerly anticipate further discussions and exchanges with you on this subject.

Comments on the Quality of English Language

The experimental material exhibits a high scientific level; however, the presentation's deficiency in English language proficiency diminishes its informative capacity. The presentation's inaccuracies become evident from the Introduction's opening sentence, which states, "Magnetosomes were first identified as nanoparticle properties that synthesize iron oxide (Fe3O4) or iron sulfide (Fe3S4)...". The object itself cannot be identified as a property; it is also unclear who (or what) synthesizes iron species under bacterial control. As illustrated in line 136 (Analytical droplet on a double-layer membrane for study.), there are instances of inconsistent sentence structure, characterized by the absence of a verb (predicate) in the sentence. To ensure the coherence and logical soundness of the text, a thorough review by the authors is imperative, accompanied by the rectification of all logical and syntactic errors. The optimal solution would be to have the text proofread by a native English speaker!

Response:Thank you for your valuable comments. We have meticulously revised all logical and grammatical errors in the manuscript and have engaged a native English speaker to assist in conducting a comprehensive linguistic review of the entire text. All revisions of English have been marked green in the revised manuscript.

Reviewer 2 Report

Comments and Suggestions for Authors

Dear Authors,

The manuscript titled: “Biocompatibility research of magnetosomes synthesized by Acidithiobacillus ferrooxidans” was written well. However, it lacks novelty in terms of characterization. The authors are required to revise the conclusion to include specific details about their findings. Here are my major comments regarding the article;

  1. The authors have used commercially available Fe3O4 magnetic nanoparticles (MNPs) for their control study. The preparation of bacterial magnetosome nanoparticles is more appropriate to write in the method section.

  1. The authors have utilized HRTEM and energy-dispersive X-ray spectroscopy (EDX) to Characterize the crystal structure and chemical composition of the magnetosomes. In addition to HRTEM and elemental mapping analysis, it is recommended that the authors provide EDAX data and a spectrum showing the peaks from the Fe and O elements.

  1. The authors performed TEM observations of magnetosomes synthesized from ferrooxidans and determined that each cell contains 10±2 magnetic nanoparticles, the particle size of about 40±6nm (Figure 2.A, B). How was the nanoparticle's size determined from the TEM? It is unclear. Please clarify. Could you please provide low-magnification TEM images of samples that show the distribution of nanoparticles?

  1. Please conduct the XPS analysis of the magnetosome synthesized from Acidithiobacillus ferrooxidans bacteria and the control Fe-oxide nanoparticles. Provide the XPS survey scan data and the percentage elemental composition tables for both. Include narrow scans for Fe-2p and O-1s for both magnetosome and the control oxide. Calculate the binding energy (BE) values difference between Fe2p peaks (Fe2p1/2 and Fe2p3/2) and compare this difference with the energy difference between the two Fe2p XPS peaks in the Fe3O4 sample. If the y matches, it confirms that the Fe-oxide present in the synthesized magnetosome is indeed Fe3O4. Also, provide the reference literature.

  1. Furthermore, it is important to demonstrate the chemical bonding between the metal and oxygen in any metal oxide nanoparticle synthesis. Therefore, FTIR characterization is necessary in this context. Please perform the FTIR for your samples to check the presence of Fe-O bonding.

Here are some minor comments:

  1. According to Figure 4, the bacterial growth lasted 48 hours instead of 40 hours.

  1. The scale bar in TEM Figure 2A is unclear. TEM images should include the magnification at which they were taken, but this information is missing.

  1. In Figure 2, captions C and D should be modified by removing the word "spectrum" and writing "Fe elemental mapping" and "O elemental mapping images."

  1. There is a repeated sentence between lines 100 and 107. Please review and make the necessary changes to the paragraph.

“This study has important academic significance for the biosafety research of bacterial magnetosomes ……. This study will promote the clinical understanding and application of magnetosomes, which ……. for the biosafety research of bacterial magnetosomes, and provides an important theoretical basis for the large-scale application of brain metastases as functional magnetic nanomaterials.”

  1. There is a spelling error in line 17 of the abstract.

  1. Please check the sentence in lines 135 and 136, page 3 of 21

Author Response

Response to Reviewer 2 Comments

  1. Summary

Thank you for your comments. Through careful thinking, we try our best to answer your suggestions and comments. All revisions of comments and suggestions have been marked red in the revised manuscript.

  1. Questions of General Evaluation

Open Review

( ) I would not like to sign my review report

(x) I would like to sign my review report

Quality of English Language

( ) The English could be improved to more clearly express the research.

(x) The English is fine and does not require any improvement.

Yes Can be improved Must be improved Not applicable

Does the introduction provide sufficient background and include all relevant references?

(x) ( ) ( ) ( )

Is the research design appropriate?

( ) ( ) (x) ( )

Are the methods adequately described?

( ) ( ) (x) ( )

Are the results clearly presented?

( ) ( ) (x) ( )

Are the conclusions supported by the results?

( ) ( ) (x) ( )

  1. Point by point response to Comments and Suggestions for Authors

Comment 1: The authors have used commercially available Fe3O4 magnetic nanoparticles (MNPs) for their control study. The preparation of bacterial magnetosome nanoparticles is more appropriate to write in the method section

Response 1: Thanks for your constructive comments. The preparation of bacterial magnetosome has been added into the Materials and methods 2.1 in revised manuscript.

Comment 2: The authors have utilized HRTEM and energy-dispersive X-ray spectroscopy (EDX) to Characterize the crystal structure and chemical composition of the magnetosomes. In addition to HRTEM and elemental mapping analysis, it is recommended that the authors provide EDAX data and a spectrum showing the peaks from the Fe and O elements.

Response 2: Thanks very much for your suggestions. We have added the EDAX data and spectrum showing  the peaks from the Fe and O elements into Results and discussion 3.2 in new revised manuscript.

Comment 3: The authors performed TEM observations of magnetosomes synthesized from ferrooxidans and determined that each cell contains 10±2 magnetic nanoparticles, the particle size of about 40±6nm (Figure 2.A, B). How was the nanoparticle's size determined from the TEM? It is unclear. Please clarify. Could you please provide low-magnification TEM images of samples that show the distribution of nanoparticles?

Response 3: Thanks for your constructive comments. Regarding the statement of “each cell contains 10±2 magnetic nanoparticles, the particle size of about 40±6nm (Figure 2.A, B)”, the low-resolution TEM images have now been added to Figure 2 in new revised manuscript. Additionally, we have removed the statement regarding “the particle size of about 40±6nm”, as the particle size distribution of the magnetosomes is primarily described in Results 3.3.

Comment 4: Please conduct the XPS analysis of the magnetosome synthesized from Acidithiobacillus ferrooxidans bacteria and the control Fe-oxide nanoparticles. Provide the XPS survey scan data and the percentage elemental composition tables for both. Include narrow scans for Fe-2p and O-1s for both magnetosome and the control oxide. Calculate the binding energy (BE) values difference between Fe2p peaks (Fe2p1/2 and Fe2p3/2) and compare this difference with the energy difference between the two Fe2p XPS peaks in the Fe3O4 sample. If the y matches, it confirms that the Fe-oxide present in the synthesized magnetosome is indeed Fe3O4. Also, provide the reference literature.

Response 4: Thanks for your comment and valuable advice. We agreed with your comment that XPS is important in confirming that the Fe-oxide present in the synthesized magnetosome is indeed Fe3O4. However, XPS testing has certain requirements for the samples, while A. ferrooxidans grows slowly and the extraction rate of magnetosomes is low, so it is difficult for the laboratory to complete the detection work in a short period of time. We have previously conducted XPS analysis on magnetosomes, but we were unable to obtain effective characteristic peaks for Fe and O. The reasons for this may be as follows: XPS primarily detects elemental information within a depth of approximately 10 nm from the sample surface. Based on the electron microscopy results presented in this paper, the crystal morphology of the magnetosomes is irregular, and the elemental distribution is likely heterogeneous, which may have prevented the detection of effective characteristic peaks. Meanwhile in the current research concerning magnetosome synthesis within magnetotactic bacteria and A. ferrooxidans, we have not yet seen related reports. Additionally, We conducted high-resolution electron microscopy and energy-dispersive spectroscopy analyses on the magnetosomes. In conjunction with prior research from our group and literature reports, we have conclusively demonstrated that the magnetosomes are composed of Fe3O4 in terms of their phase composition. We will pay attention to such problems in our future study. Hope this is acceptable.

Comment 5: Furthermore, it is important to demonstrate the chemical bonding between the metal and oxygen in any metal oxide nanoparticle synthesis. Therefore, FTIR characterization is necessary in this context. Please perform the FTIR for your samples to check the presence of Fe-O bonding.

Response 5: Thanks very much for your suggestions. We agreed with your comment that FTIR characterization is necessary in this context. However, due to the slow growth of A. ferrooxidans and the relatively low extraction yield of magnetosomes, it is really difficult for us to complete these preparations and analysis within a short time. Additionally, in present, it is documented in published studies that the magnetosomes synthesized by Acidithiobacillus ferrooxidans are composed of Fe3O4 crystals. The following literature is the related report we provide to you. Meanwhile, in this study, we conducted high-resolution electron microscopy and energy-dispersive spectroscopy (EDS) analysis on the magnetosomes. The EDS results confirmed that the elemental composition primarily consists of Fe and O. Furthermore, the lattice fringes observed in the high-resolution images provide evidence that the crystals are Fe3O4. We will pay attention to such problems in our future study. Hope this is acceptable. 

  • Lei Yan, Yue X , Zhang S ,et al.Biocompatibility evaluation of magnetosomes formed by Acidithiobacillus ferrooxidans[J].Materials Science and Engineering: C, 2012.DOI:10.1016/j.msec.2012.04.062.
  • Zhao D , Yang J , Zhang G ,et al.Potential and whole-genome sequence-based mechanism of elongated-prismatic magnetite magnetosome formation in Acidithiobacillus ferrooxidansBYM[J].World Journal of Microbiology & Biotechnology, 2022(7):38.DOI:10.1007/s11274-022-03308-2.
  • Wu L , Yang B , Wang X ,et al.Effects of Single and Mixed Energy Sources on Intracellular Nanoparticles Synthesized by Acidithiobacillus ferrooxidans[J].Minerals, 2019, 9(3):163.DOI:10.3390/min9030163.

Minor comments:

Thanks for your minor comments. We have corrected all the errors refereed to your comments in new revised manuscript.

  1. According to Figure 4, the bacterial growth lasted 48 hours instead of 40 hours.

Response 6: According to your suggestion, We have corrected it.

  1. The scale bar in TEM Figure 2A is unclear. TEM images should include the magnification at which they were taken, but this information is missing.

Response 7: We have revised Figure 2 to enhance the clarity of the scale bar.

  1. In Figure 2, captions C and D should be modified by removing the word "spectrum" and writing "Fe elemental mapping" and "O elemental mapping images."

Response 8: Caption C and D in Figure 2 have been corrected according to your comment.

  1. There is a repeated sentence between lines 100 and 107. Please review and make the necessary changes to the paragraph.

Response 9: We have deleted the repeated sentence in Line 104-106.

  1. There is a spelling error in line 17 of the abstract.

Response 10: This spelling error “degration” has been corrected to “degradation” in Line 18.

  1. Please check the sentence in lines 135 and 136, page 3 of 21

Response 11: Thanks for your reminder. This sentence has been corrected in Lines 146 -148.

Reviewer 3 Report

Comments and Suggestions for Authors

The manuscript addresses a broad biological characterization of magnetosomes produced by bacteria (Acidithiobacillus ferrooxidans). The context of the research is relevant, and the results are of great interest to the field. The methods were well described, as were the results. I consider the paper suitable for publication. However, a series of modifications and corrections are needed, especially regarding the writing quality. Below are my comments, which I hope will assist the authors in improving their manuscript.

Line 25: Does the paper provide a theoretical or experimental basis?

Line 30: Revise "identified as nanoparticles properties"

Line Line 34: Specify the abbreviation "BMs", since it is the first time that it is used.

Line 38: Please, clarify why "Because of its inherent magnetic properties, targeted drug delivery is mitigated". It is not clear for me.

Line 50 : Starting with "Among various...". Please, revise to avoid repetitions.

Line 67: Please, discuss what was observed in the mentioned study.

Line 76: About "...is also a Gram-negative...". This characteristic was not discussed previously for other bacteria. Please, revise.

Line 83: Please, revise the capital letter "A" in the middle of the text.

Line 99: Use the abbreviation "BMs", since it is already defined before.

Line 117: Revise "solution". The system consists in a suspension, right?

Line 175: Revise "respectively". 

Lines 183-186: Please, stardadize the verbal tense (past?).

Line 205: Why the BMs concentration was not standardize (in some cases 4 mg mL-1 is used, in another cases 6)?

Figure 1 could be in SI.

Line 252-256: Please, standardize the verbal tense. Revise for improving readability.

Have the authors checked the visual stability of the suspension? It is observed fases separation along time?

Line 288: Should 30.9 mV be actually -30.9 mV?

Line 301: Please, revise the provided values in terms of significant algaritms, here and elsewhere in the manuscript. For instance, 21.95 +/- 2.28 should be expressed as 22 +/- 2.

Line 325: Mentioning the orgas again is repetitive. Please, revise.

In the caption of Figure 6, please, mention what the arrows point to. 

Please, revise the paragraph starting in 435 to improve clarity.

Line 455: Instead of mentioning that a certain result was different, please, mention if a lower or higher value was observed.

Line 478: Please, revise "The main source of A. ferrooxidans are iron(II)".

Line 500: Please explain why is amorphous is worse. In which sense?

Comments on the Quality of English Language

Line 14: Should "intracellur" be substituted by "intracellular"?

Line 21: Please, consider "it didn't cause" instead of "it would not cause".

Line 284: Please consider, for instance, "showed" instead of "found".

Author Response

Response to Reviewer 3 Comments

  1. Summary

Thank you for your comments. Through careful thinking, we try our best to answer your suggestions and comments. All revisions of comments have been corrected and marked red in the revised manuscript.

  1. Questions of General Evaluation

Open Review

(x) I would not like to sign my review report

( ) I would like to sign my review report

Quality of English Language

(x) The English could be improved to more clearly express the research.

( ) The English is fine and does not require any improvement.

Yes Can be improved Must be improved Not applicable

Does the introduction provide sufficient background and include all relevant references?

(x) ( ) ( ) ( )

Is the research design appropriate?

(x) ( ) ( ) ( )

Are the methods adequately described?

(x) ( ) ( ) ( )

Are the results clearly presented?

(x) ( ) ( ) ( )

Are the conclusions supported by the results?

(x) ( ) ( ) ( )

  1. Point by point response to Comments and Suggestions for Authors

Comment 1 Line 25: Does the paper provide a theoretical or experimental basis?

Re. 1: Yes, this paper provides a theoretical and experimental basis for subsequent large-scale applications of magnetosomes in biomedical fields.

Comment 2 Line 30: Revise "identified as nanoparticles properties"

Re. 2: According to your suggestions, this sentence has been corrected in Line 30 of revised manuscript .

Comment 3  Line 34: Specify the abbreviation "BMs", since it is the first time that it is used.

Re. 3: According to your suggestions, this error has been corrected in Line 34.

Comment 4 Line 38: Please, clarify why "Because of its inherent magnetic properties, targeted drug delivery is mitigated". It is not clear for me.

Re. 4: This sentence has been revised to “The intrinsic magnetic properties facilitate the targeted drug delivery process” in Line 39-39 of revised manuscript.

Comment 5 Line 50: Starting with "Among various...". Please, revise to avoid repetitions.

Re. 5: We have deleted the repeated sentence in revised manuscript

Comment 6 Line 67 Please, discuss what was observed in the mentioned study.

Re. 6: The discuss of mentioned study has been added into Introduction Line 67-68.

Comment 7 Line 76: About "...is also a Gram-negative...". This characteristic was not discussed previously for other bacteria. Please, revise.

Re. 7: The “A. ferrooxidans is also a Gram-negative autotrophic rod-like bacteria.”has been revised to “Meanwhile, A. ferrooxidans is a kind of Gram-negative autotrophic rod-like bacteria” in Line 76.

Comment 8 Line 83: Please, revise the capital letter "A" in the middle of the text.

Re. 8: The capital letter ”A” has been corrected to “a” in Line 84.

Comment 9 Line 99: Use the abbreviation "BMs", since it is already defined before.

Re. 9: The “bacterial magnetosomes(BMs for short)” has been revised to “BMs” in Line 99.

Comment 10 Line 117: Revise "solution". The system consists in a suspension, right?

Re. 10: Yes, the “solution” has been corrected to “suspension” in Line 123.

Comment 11 Line 175: Revise "respectively".

Re. 11: We have deleted “respectively” in Line 182. 

Comment 12 Lines 183-186: Please, stardadize the verbal tense (past?).

Re. 12: Yes, it would be past. The verbal are all revised in 190-193.

Comment 13 Why the BMs concentration was not standardize (in some cases 4 mg mL-1 is used, in another cases 6)?

Re. 13: The concentration of injected BMs will be adjusted according to the specific requirements of the biocompatibility study, enabling a comprehensive and objective evaluation of the biocompatibility of BMs from multiple perspectives. For example, injection concentrations of rat and mouse would be different.

Comment 14 Figure 1 could be in SI.

Re. 14: Thanks for your suggestion sincerely. We considered about your advice carefully.In this paper, magnetosomes are autonomously mineralized and synthesized intracellularly by A. ferrooxidans. The growth status of A. ferrooxidans. is particularly crucial. Therefore, we believe that the growth curve of the bacteria is key data and should be included in the main body of the paper.

Comment 15 Line 252-256: Please, standardize the verbal tense. Revise for improving readability.

Re. 15: The verbal tense has been standardized according to your suggestion in 260-270.

Comment 16 Have the authors checked the visual stability of the suspension? It is observed fases separation along time?

Re. 16: We have checked the visual stability of the suspension during the whole process of experiment. There is no phases separation along time. The suspension is stable.

Comment 17 Line 288: Should 30.9 mV be actually -30.9 mV?

Re. 17: Yes, it is -30.9mV. We have corrected it in Line 297.

Comment 18 Line 301 Please, revise the provided values in terms of significant algaritms, here and elsewhere in the manuscript. For instance, 21.95 +/- 2.28 should be expressed as 22 +/- 2

Re. 18: We have revised the values in terms of significant algaritms in whole newly revised manuscript according to your comments and guides.

Comment 19 Line 325: Mentioning the orgas again is repetitive. Please, revise.

Re. 19: We have deleted “the major accumulating organs of” in Line 335.

Comment 20 In the caption of Figure 6, please, mention what the arrows point to.

Re. 20: We have added the explanation about what the arrows point to in caption of Figure 6.

Commen 21 Please, revise the paragraph starting in 435 to improve clarity.

Re. 21: We have revise the paragraph starting in Line 445 to improve clarity .

Comment 22 Line 455: Instead of mentioning that a certain result was different, please, mention if a lower or higher value was observed.

Re. 22: We have corrected that a lower value was observed.

Comment 23 Line 478: Please, revise "The main source of A. ferrooxidans are iron(II)".

Re. 23: We have revise this to “The main source of A. ferrooxidans are ferrite and minerals” in Line 487.

Comment 24 Line 500: Please explain why is amorphous is worse. In which sense?

Re. 24: According to your suggestion, we have considered and deleted the sentence “which is slightly worse than that of magnetotactic bacteria” in Line 503 .

Round 2

Reviewer 2 Report

Comments and Suggestions for Authors

Thank you for submitting the revised manuscript titled "Biocompatibility Research of Magnetosomes Synthesized by 2 Acidithiobacillus Ferrooxidans." I am satisfied with the responses to my comments 1, 2, and 3. The authors have addressed these comments with additional details regarding methods and materials, including TEM and EDS data, further clarification, and low-magnification TEM images in Figure 2. However, regarding my comments 4 and 5, the authors have not provided the characterization data from FTIR and XPS. These experiments should be performed as a part of material characterization. I suggest the author provide prior research conducted by their group where they have established that the magnetosomes consist of the Fe3O4 phase. While the authors have provided three references to relate their work to those referenced, I found the XRD characterization from these references to be crucial, and it is missing from the authors' revised version. If possible, I recommend conducting this analysis. Finally, the authors have adequately addressed all minor comments. The revised version shows improvement; however, due to insufficient data to confirm the successful synthesis of magnetosomes, further revision is necessary. Thank you!

Author Response

Response to Reviewer 2 Comments

  1. Summary

Thank you for your comments. According to your suggestions, we have made careful revisions. All revisions of comments and suggestions have been marked red in the revised manuscript. 

  1. Comments and Suggestions for Authors

Thank you for submitting the revised manuscript titled "Biocompatibility Research of Magnetosomes Synthesized by 2 Acidithiobacillus Ferrooxidans." I am satisfied with the responses to my comments 1, 2, and 3. The authors have addressed these comments with additional details regarding methods and materials, including TEM and EDS data, further clarification, and low-magnification TEM images in Figure 2. However, regarding my comments 4 and 5, the authors have not provided the characterization data from FTIR and XPS. These experiments should be performed as a part of material characterization. I suggest the author provide prior research conducted by their group where they have established that the magnetosomes consist of the Fe3O4 phase. While the authors have provided three references to relate their work to those referenced, I found the XRD characterization from these references to be crucial, and it is missing from the authors' revised version. If possible, I recommend conducting this analysis. Finally, the authors have adequately addressed all minor comments. The revised version shows improvement; however, due to insufficient data to confirm the successful synthesis of magnetosomes, further revision is necessary. Thank you!

Response: Thanks for your comment and valuable advice. We agreed with your comment that XRD characterization is crucial. Therefore, we have supplemented the  method, results, and discussion of XRD analysis into the revised manuscript according to your suggestions, with the modifications highlighted in red. The results confirm that the magnetosomes we prepared exhibit a phase composition of Fe3O4. Additionally, we have provided prior research conducted by our group where we have established that the magnetosomes consist of the Fe3O4 phase for your reference. We hope that these revisions and responses meet your expectations.

The following content is excerpted from the 2019 paper by Dr. Lingbo Wu from our research group, published in Minerals (title: Effects of Single and Mixed Energy Sources on Intracellular Nanoparticles Synthesized by Acidithiobacillus ferrooxidans). Notably, the Acidithiobacillus ferrooxidans strain used in the cited study is identical to that employed in the present manuscript ijms-3523181[1].

  1. The previous work of this part of the research group mainly investigated the effects of single and mixed energy sources on intracellular nanoparticles synthesized by Acidithiobacillus ferrooxidans. We carried out high-resolution transmission electron microscopy characterization of the magnetosomes synthesized by Acidithiobacillus ferrooxidansunder three different mixed energy ratios (FeSO47H2O:S = 0.16: 0, 0.16: 0.25, 0.16: 0.5 mol/L). The energy ratio FeSO4●7H2O:S = 0.16: 0, which represent 44.7g/L FeSO4●7H2O, was consistent with the present manuscript ijms-3523181. as shown in Figure 1-1, the lattice spacing of magnetosomes cultured in the ratio 1#, 2#, and 3# was 2.95 Å, 2.53 Å, and 2.43 Å, respectively. The fringes with spacing was clearly corresponded to magnetite.
  2. Fe K-edge XANES analyses were carried out.As shown in Figure 2-2, the spectra of standard Fe-containing compounds showed differences in the peaks' When we examined the XANES spectra of 1#,2#,and 3#,we observed a shoulder feature at 7139 eV (green dotted line,Figure 2-2),which was consistent with Fe3O4. The spectra of Fe2O3 and Fe(OH)3 had two peaks at 7129 and 7134eV(red arrow,Figure 2-2).The spectra of Fe3O4 had one peak at 7131 eV (black arrow, Figure 2-2). Moreover,the spectra of nanoparticles cultured in the media 1#,2#,and 3#had strong peaks at 7131eV and had no peak at 7134eV. Therefore,the spectra of 1#to 3#were more similar to Fe3O4 than Fe(OH)3 and Fe2O3. Overall, by combining the results of HRTEM,and XANES, it could be proved that the composition of the magnetosomes in Acidithiobacillus ferrooxidans was magnetite

  1. L. Wu, B. Yang, X. Wang, et al., Effects of Single and Mixed Energy Sources on Intracellular Nanoparticles Synthesized by Acidithiobacillus ferrooxidans [J]. Minerals, 2019. 9(3). https://doi.org/10.3390/min9030163:

Figure 1-1 High-resolution transmission electron microscopy (HRTEM) analysis of the nanoparticles cultured in media 1# (a), 2# (b), and 3# (c).

Figure 2-2 X-ray absorption near edge structure (XANES) spectra of bacteria cultured in media 1#(a), 2#(b), and 3#(c).

Round 3

Reviewer 2 Report

Comments and Suggestions for Authors

The authors have performed XRD characterization of the synthesized magnetosomes and demonstrated that the phase evolution corresponds to that of Fe2O3 NPs. They have provided the reference [27] as evidence, and it aligns with the XRD data of their synthesized sample. Additionally, authors have provided an explanation for this based on their prior research, which meets my expectations.

My recommendation is that the manuscript, in its present form, is well-revised and very suitable for publication in a journal. I have no further comments on the manuscript. 

Thank you for submitting the revised manuscript.

Author Response

Response to Academic Editor

Thanks for your comments and recommendation. We sincerely appreciate your positive feedback on our paper and are deeply grateful for the valuable suggestions you previously provided. We look forward to potential future opportunities for further academic exchanges with you. Wishing you all the best in your endeavors.
